# Learning the distribution of single-cell chromosome conformations in bacteria reveals emergent order across genomic scales

Joris J. B. Messelink[1], Muriel C. F. van Teeseling [2,6], Jacqueline Janssen[1], Martin Thanbichler [2,3,4] & Chase P. Broedersz [1,5✉]

The order and variability of bacterial chromosome organization, contained within the distribution of chromosome conformations, are unclear. Here, we develop a fully data-driven maximum entropy approach to extract single-cell 3D chromosome conformations from Hi–C experiments on the model organism *Caulobacter crescentus*. The predictive power of our model is validated by independent experiments. We find that on large genomic scales, organizational features are predominantly present along the long cell axis: chromosomal loci exhibit striking long-ranged two-point axial correlations, indicating emergent order. This organization is associated with large genomic clusters we term Super Domains (SuDs), whose existence we support with super-resolution microscopy. On smaller genomic scales, our model reveals chromosome extensions that correlate with transcriptional and loop extrusion activity. Finally, we quantify the information contained in chromosome organization that may guide cellular processes. Our approach can be extended to other species, providing a general strategy to resolve variability in single-cell chromosomal organization.

[1] Arnold Sommerfeld Center for Theoretical Physics and Center for NanoScience, Department of Physics, Ludwig Maximilian University Munich, Munich, Germany. [2] Department of Biology, University of Marburg, Marburg, Germany. [3] Max Planck Institute for Terrestrial Microbiology, Marburg, Germany. [4] Center for Synthetic Microbiology (SYNMIKRO), Marburg, Germany. [5] Department of Physics and Astronomy, Vrije Universiteit Amsterdam, Amsterdam, The Netherlands. [6] Present address: Prokaryotic Cell Biology Group, Department of Microbial Interactions, Institute for Microbiology, Friedrich Schiller University Jena, Jena, Germany. ✉email: c.broedersz@lmu.de

Chromosomes carry all information to generate a living cell. In both prokaryotes and eukaryotes, chromosomal DNA is highly compacted to fit inside its cellular confinement. This implies a major organizational problem: the DNA does not only have to be highly condensed, but its spatial organization also has to facilitate processes such as transcription and replication. In many bacteria, the genetic information is stored on a single chromosome with a contour length three orders of magnitude larger than the cell. Various proteins regulate bacterial chromosome structure[1–5], imposing order on its spatial organization and thereby impacting cellular processes such as transcription[6]. However, this order is opposed by thermal[7] and active chromosomal fluctuations[8], as well as inherent cell-to-cell variability[9]. The resulting degree of organization of the chromosome remains unclear. Resolving this organization requires a characterization of the distribution of single-cell chromosome conformations, posing a key challenge for experiment and theory[10].

The classical picture in which the bacterial chromosome is arranged as an amorphous polymer has become obsolete thanks to recent experimental advances[11–13]. Indeed, fluorescence microscopy experiments revealed that chromosomal loci localize to well-defined cellular addresses in various species[7,14–16], including *Caulobacter crescentus*[17]. This organization helps steer chromosome segregation[18] and cell division[19]. In addition, the level of transcription of several genes depends on their distance to the pole[20]. Further insights were obtained by chromosome conformation capture 5C/Hi–C experiments[21,22], measuring average pair-wise contacts between loci. These experiments revealed Chromosomal Interaction Domains (CIDs) of up to $10^5$ base pairs, comprising loci preferentially interacting within their domain. Various processes[23,24], including transcription[25,26], impact CID organization. On larger genomic scales, locus pairs on opposite chromosomal arms appear to favor a juxtaposed arrangement in several species, induced by the loop extrusion motor SMC (Structural Maintenance of Chromosomes)[23,26–31]. However, it remains challenging to faithfully extract the distribution of 3D chromosome conformations from Hi–C data. Thus, despite these experimental insights, a complete model for the spatial organization of the bacterial chromosome across genomic scales remains elusive.

To exploit advances in Hi–C experiments on various bacteria[23,24,26,29,31,32], a principled data-driven approach is needed that makes an unbiased inference of the distribution of chromosome configurations. However, there are several outstanding challenges that preclude such a fully data-driven model[26,27,33,34]. Several approaches rely on an assumed relation between Hi–C scores and the average spatial distance between locus pairs to obtain a 3D structure[27,33,35]. Other approaches generate an ensemble of configurations consistent with Hi–C data, e.g., using iterative maximum likelihood algorithms[36]. However, Hi–C maps could be consistent with many underlying distributions. For eukaryotes, an equilibrium Maximum Entropy (MaxEnt) distribution selection method was proposed[37–39], as used for protein structure prediction[40]. However, such an approach may be unsuitable for chromosomes in living cells, which exhibit non-equilibrium fluctuations[8,41,42]. Thus, a rigorous approach to derive a distribution of chromosome conformations compatible with non-equilibrium dynamics is still lacking.

Here, we develop a fully data-driven MaxEnt approach for the bacterial chromosome based on Hi–C data. This approach infers the least-structured distribution of chromosome conformations that fits Hi–C experiments, capturing population heterogeneity at the single-cell level. Our MaxEnt model does not rely on equilibrium assumptions, is inferred directly from normalized Hi–C scores, does not require an assumed Hi–C score-distance relation,

and we determine the coarse-graining scale of our model using experiments. The MaxEnt model reveals the organization and variability of the bacterial chromosome across genomic scales. Using this model, we quantify the localization information in the cellular location of chromosomal loci that can be used by cellular processes. Our theoretical framework may be generalized to other prokaryotic and eukaryotic species, providing a principled approach to resolve chromosome organization from Hi–C data.

## Results

### Maximum entropy model inferred from chromosomal contact frequencies.

Our goal is to determine the ensemble of single-cell chromosome conformations for a heterogeneous cell population from experimental Hi–C data. To this end, we build on existing MaxEnt methods for analyzing biophysical data[37,38,40,43–49], to develop a principled approach for inferring the statistics of chromosome structure in bacteria from experiments.

The microstates $\{\sigma\}$ of the system are defined as the set of all configurations of the chromosome contained within the cellular confinement. We seek the statistical weights $P(\sigma)$, chosen to be consistent with the experimental Hi–C map. In general, however, a set of experimental constraints does not uniquely determine $P(\sigma)$. The MaxEnt approach is based on selecting $P(\sigma)$ from these possible solutions by choosing the unique distribution with the largest Shannon entropy,

$$S = -\sum_{\sigma} P(\sigma)\ln P(\sigma), \qquad (1)$$

constituting the least-structured distribution consistent with experimental data. Put simply, we require that the only structure present in $P(\sigma)$ is due to experimental constraints from Hi–C scores, rather than assumed features of the underlying polymer model, the interpretation of Hi–C scores, or the ensemble-generating algorithm. A central assumption of our approach is that the experimental Hi–C maps contain sufficient information to constrain the distribution of chromosome conformations.

To apply the MaxEnt method to experimental Hi–C data, we employ a coarse-grained representation of the chromosome: the polymer is represented as a discrete circular chain of length $N$ on a 3D cubic lattice; the chain can self-intersect and is constrained to the cell-shaped confinement. A subset of the $N$ monomers—equally spaced along this chain—represents the centers of the genomic regions, which are defined as the stretch of the DNA associated with an individual bin of the Hi–C map. Thus, the dimensions of the coarse-grained representation are set by the resolution of the available Hi–C data (Supplementary Notes 2, 3.1). This provides an efficient computational framework, while still capturing key organizational features. Specifically, this representation is chosen to preserve experimentally measured distance fluctuations at the coarse-graining scale (see "Methods" section and Supplementary Notes 1–2). At larger scales, the statistics of polymer configurations are only constrained by Hi–C data. Within this representation, a microstate $\sigma = \{\mathbf{r}_1, \mathbf{r}_2, \dots\} = \{\mathbf{r}\}$ is defined by the monomer positions $\mathbf{r}_i$. Two genomic regions have a contact probability $\gamma$ if they occupy the same lattice site, and 0 otherwise.

To obtain the least-structured distribution of microstates consistent with experiments, we seek $P(\{\mathbf{r}\})$ that maximizes $S$ (Eq. (1)) under experimental constraints[45,50]. The two constraints we impose are: 1) the model contact frequencies should match experimental contact frequencies $f_{ij}^{\text{expt}}$ between genomic regions $i$ and $j$ (the correspondence between $f_{ij}^{\text{expt}}$ and Hi–C scores is discussed in the next section), and 2) the distribution should be normalized. To this end, we introduce the functional $\tilde{S}$, with one Lagrange multiplier $\lambda_{ij}$ for each experimental constraint and $\lambda_0$

ensuring normalization:

$$\tilde{S} = - \sum_{\{\mathbf{r}\}} P(\{\mathbf{r}\}) \ln P(\{\mathbf{r}\}) - \sum_{ij} \lambda_{ij} \left( \sum_{\{\mathbf{r}\}} P(\{\mathbf{r}\}) \gamma \delta_{\mathbf{r}_i, \mathbf{r}_j} \right.$$
$$\left. - f_{ij}^{\text{expt}} \right) - \lambda_0 \left( \sum_{\{\mathbf{r}\}} P(\{\mathbf{r}\}) - 1 \right) \tag{2}$$

Here, $\delta_{\mathbf{r}_i, \mathbf{r}_j}$ is the Kronecker delta. We maximize $\tilde{S}$ under these constraints, setting $\frac{\delta \tilde{S}}{\delta P(\{\mathbf{r}\})} = 0$, yielding

$$P(\{\mathbf{r}\}) = \frac{1}{Z} \exp \left[ -\sum_{ij} \lambda_{ij} \gamma \delta_{\mathbf{r}_i, \mathbf{r}_j} \right], \tag{3}$$

with $Z = \exp[1 + \lambda_0]$. The $\lambda_{ij}$'s parametrizing $P(\{\mathbf{r}\})$ is determined by solving

$$\sum_{\{\mathbf{r}\}} P(\{\mathbf{r}\}) \gamma \delta_{\mathbf{r}_i, \mathbf{r}_j} = f_{ij}^{\text{expt}} \tag{4}$$

for each experimental constraint. For typical Hi–C data on a bacterial chromosome, this amounts to of order $10^5$ constraints[26]. These equations can not be solved directly, as they are highly nonlinear and the state space is very large.

The daunting challenge of finding the Lagrange multipliers can be overcome by noting that the distribution in Eq. (3) can be mapped to a statistical mechanics model: a confined lattice polymer, with a (dimensionless) Hamiltonian

$$H = \frac{1}{2} \sum_{ij} \epsilon_{ij} \delta_{\mathbf{r}_i, \mathbf{r}_j}. \tag{5}$$

The mapping to Eq. (3) is made by setting $\epsilon_{ij} = \gamma \lambda_{ij}$, where $\epsilon_{ij}$ are the effective interaction energies between overlapping loci in the Hamiltonian formulation. Importantly, although a mapping can be made to a statistical mechanics model, our approach does not rely on the chromosome being in thermal equilibrium. This is in contrast to approaches used in refs. [37–39] where a hybrid MaxEnt procedure is employed combining a physical polymer model with Hi–C derived constraints, resulting in an energy landscape description of equilibrium chromosome configurations.

We numerically obtain the inverse solutions of this model using iterative Monte Carlo simulations (Supplementary Note 3). Testing this algorithm on contact frequency maps generated from a set of chosen input $\epsilon_{ij}$, we find that our algorithm precisely and robustly recovers the correct input values (Supplementary Note 4).

**Inferring the MaxEnt model directly from normalized Hi–C scores.** A major hurdle in applying data-driven inference approaches is finding a correspondence between experimental Hi–C scores and the contact frequencies in a coarse-grained polymer model. Published Hi–C maps are typically normalized. This normalization compensates known biases in raw Hi–C data, for instance, due to the proportionality between the number of restriction sites in a genomic region and its Hi–C score[51]. Furthermore, absolute Hi–C scores are hard to interpret because it is difficult to estimate the conversion factor to physical contact frequencies. Importantly, however, even if absolute contact scores could be obtained, a mapping to contact frequencies in a coarse-grained model is challenging.

We address this conversion issue by treating the conversion factor as an unknown parameter $c$ in our MaxEnt procedure. Thus, we write $f_{ij}^{\text{expt}} = c \tilde{f}_{ij}^{\text{expt}}$, with $\tilde{f}_{ij}^{\text{expt}}$ the normalized experimental Hi–C scores. We absorb the contact probability factor $\gamma$ into $c$ (Eq. (2)), setting $\tilde{c} = \frac{c}{\gamma}$, and require that $\tilde{c}$ maximizes the model entropy (Supplementary Note 3.2), yielding the additional

constraint

$$\sum_{ij} \epsilon_{ij} \tilde{f}_{ij}^{\text{expt}} = 0. \tag{6}$$

Thus, we infer the least-structured distribution of chromosome conformations from normalized Hi–C data, without assuming a conversion between Hi–C scores and contact frequencies or average distances between loci.

**MaxEnt model of the *C. crescentus* chromosome quantitatively captures measured cellular localization.** We investigate the degree of organization of the bacterial chromosome by considering newborn swarmer cells of the model organism *C. crescentus*. Such newborn swarmer cells contain only a single chromosome, whose replication has not yet initiated[52]. To develop the MaxEnt model for *C. crescentus*, we first experimentally determine the coarse-graining scale, set by the average distance between consecutive 10 kb genomic regions (Supplementary Notes 1–2). Subsequently, we infer the parameters of the MaxEnt model from published experimental Hi–C data (Supplementary Note 5)[26]. Our inverse algorithm robustly converges to an accurate description of the Hi–C map: the modeled and experimental contact maps have an average pair-wise deviation of 6.0% of the total average Hi–C score with a Pearson's correlation coefficient of 0.998 (Fig. 1A, B inset).

Our MaxEnt model quantitatively reproduces essential features of the experimental Hi–C map (Fig. 1A), including the fine structure of the CIDs, as well as the secondary diagonal, which is attributed to the alignment of the two chromosomal arms by SMC[30,53–55]. The inferred $\epsilon_{ij}$'s (Fig. 1B) should not be interpreted as physical interaction energies. Rather, they parametrize the predicted physical distribution of chromosome configurations $P(\{\mathbf{r}_i\})$. We can directly interpret the organizational features implied by $P(\{\mathbf{r}_i\})$ and use it to sample single-cell configurations (Fig. 1C).

We test the predictive power of the MaxEnt model by computing the distribution of axial locations of several loci. Importantly, we do not assume (polar) cell envelope tethering of specific loci, such as the origin of replication (*ori*). We orient cells by setting the *ori* pole in the cell-half containing *ori*. Interestingly, we find a high degree of axial localization of loci: the average axial position of loci is roughly linearly organized, and the predicted positions match previous live-cell microscopy experiments[17] (Fig. 2A). By contrast, simulation results of a confined random polymer—not constrained by Hi–C data—do not exhibit the linear organization, even when *ori* is tethered to the cell pole.

The MaxEnt model also predicts distributions of long-axis positions of chromosomal loci, in remarkable agreement with prior experiments (Fig. 2B). This comparison with independent experimental data constitutes a strong validation of our MaxEnt model. The slight deviation of the position of *ori* compared to the experiments (Fig. 2A, B) can be addressed with an extended MaxEnt model that incorporates the distribution of axial *ori* positions as an additional constraint (Supplementary Note 17). However, other aspects of the predicted chromosomal organization are largely unaffected by this modification, and therefore we will not impose this additional constraint in our analysis.

**Large-scale chromosome organization primarily characterized by long-axis correlations associated with Super Domains.** Large-scale organizational features of the chromosome can be revealed by measuring various two-point correlation functions. Earlier models suggested a three-dimensional organization in which the two chromosomal arms wind around each other with roughly one helical turn[27,33]. To test if this organization also

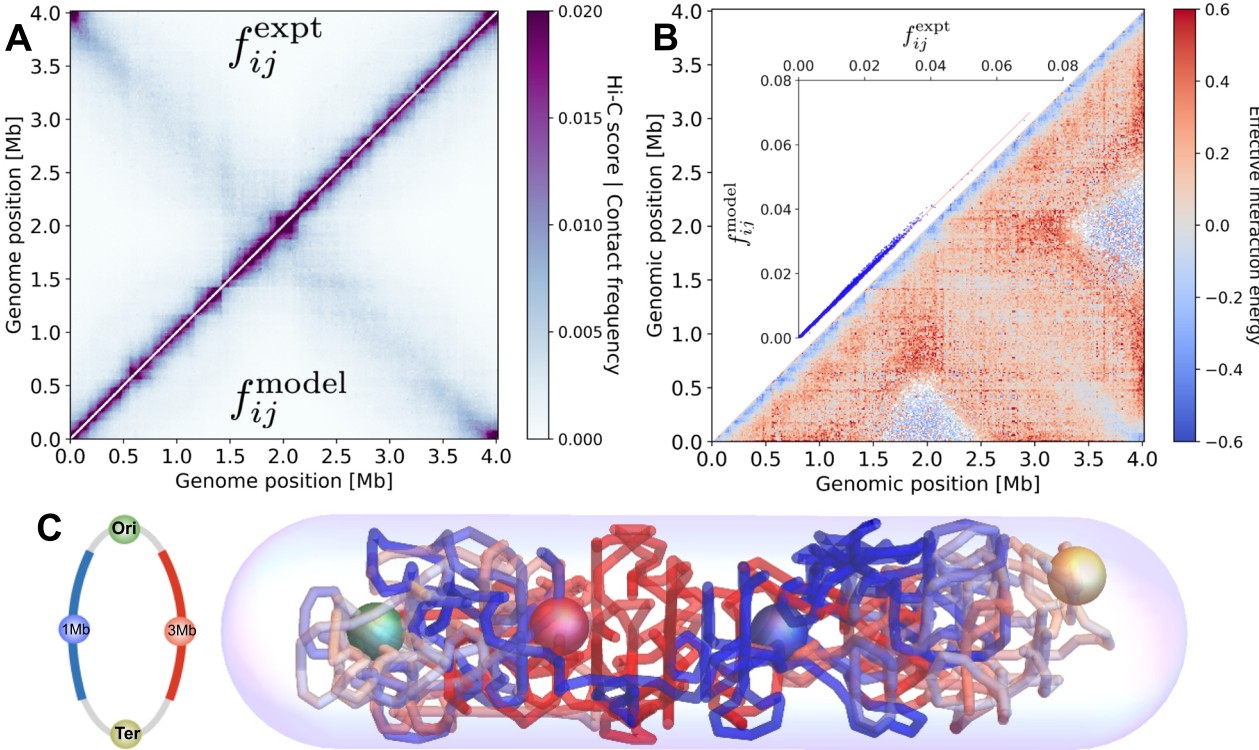

**Fig. 1 Maximum entropy model inferred from Hi–C experiments in *C. crescentus*. A** Comparison between experimental contact frequencies $f_{ij}^{\text{expt}}$ (upper left corner, adapted from ref. [26]) and contact frequencies obtained from our inferred MaxEnt model $f_{ij}^{\text{model}}$ (lower right corner). **B** Associated inferred effective interaction energies $\epsilon_{ij}$ (lower right corner, white regions indicate $\epsilon_{ij} \to \infty$) together with a scatter plot of $f_{ij}^{\text{expt}}$ vs. $f_{ij}^{\text{model}}$ (inset). **C** Visualization of a single-cell chromosome configuration predicted by our MaxEnt model; the centers of four distinct chromosome sections are represented in the schematic by colored spheres.

emerges in our MaxEnt model, we compute two-point correlations of angular orientations. For each chromosome segment, we assign an orientation vector in the plane perpendicular to the long axis. We find that angular correlations decay rapidly for genomic distances ⪆0.2 Mb (Fig. 3A lower right). Large-scale helical order is thus negligible, indicating that a pronounced helical organization is not required to model the experimental Hi–C map.

The two-point correlation function in radial positions decays even more rapidly with genomic distance up to ~0.1 Mb (Fig. 3A upper left), indicating the absence of large-scale order in this direction. By contrast, two-point correlations in the long-axis position exhibit a striking structure: we observe positive long-ranged correlations for pairs of genomic regions on the same chromosomal arm, whereas correlations in axial positions between arms are predominantly negative (Fig. 3B upper left). These long-ranged correlations signify emergent order. Importantly, such organization is absent for a model with a tethered origin not constrained by Hi–C data (Fig. 3B, lower right), as well as for a model with juxtaposed chromosomal arms only constrained by linearly organized average long-axis positions (Supplementary Note 16). Moreover, the structure of the long-axis correlations is inconsistent with global rotational fluctuations (Supplementary Note 12).

We find that these intra-arm anticorrelations are associated with large high-density clusters of subsequent genomic regions, which we term Super Domains (SuDs). SuDs emerge from a clustering analysis of genomic regions in single-cell conformations (Supplementary Note 9). The formation of domain-like structures is revealed by plotting the distance between pairs of loci for a specific chromosome configuration, with single domains spanning up to a quarter of the chromosome length (Fig. 4A, B). On average, 73% of genomic regions are part of a SuD, each

chromosomal arm contains ~4 SuDs, and each SuD contains 48 genomic regions (Supplementary Fig. 21). Compared to CIDs, they are typically larger with more variable size and genomic location across chromosome conformations. The variable and delocalized nature of SuDs is apparent from the average distance map between genomic regions, indicating no discrete structure (Fig. 4C). Importantly, SuDs forming on opposing chromosomal arms tend to spatially exclude each other (Fig. 4B, E): the fraction of overlap in axial positions is reduced by 26% compared to randomly paired left and right arm configurations. As a result of this tendency to spatially exclude, chromosomal regions belonging to SuDs on opposing sections of the two arms, are expected to fluctuate in an anti-correlated fashion. (Supplementary Note 9). Thus, this exclusion behavior of opposing SuDs is expected to generate negative intra-arm correlations for pairs of genomic regions with similar average axial positions (Supplementary Note 9).

To experimentally verify signatures of SuDs, we turned towards SIM (structured illumination microscopy) super-resolution microscopy and investigated the intracellular distribution of chromosomal DNA in *C. crescentus* at the single-cell level. These experiments reveal that the chromosome exhibits a highly heterogeneous spatial distribution in the cell, including several dense cluster-like regions (Fig. 4D). We observe that the number, size, and location of these high-density regions are found to vary from cell to cell, consistent with SuD properties derived from our MaxEnt model. To compare these single-cell experimental results with theory, we provide computed density plots of chromosomes based on our MaxEnt model. Specifically, for each chromosome configuration in our model, we compute a chromosome density plot at the experimental resolution (see Methods), as shown in (Fig. 4E). In the computed density plots, we observe high-density

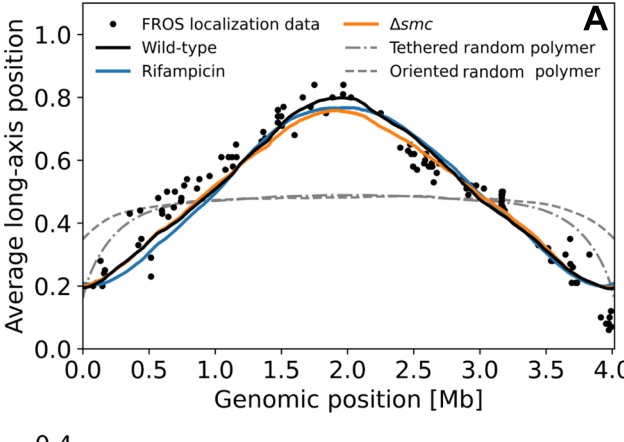

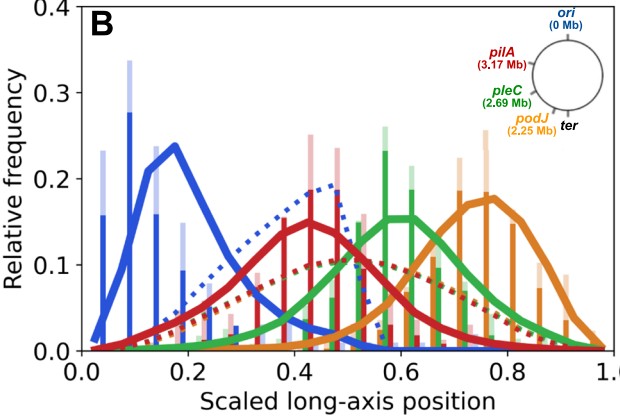

**Fig. 2 Validation of MaxEnt model based on spatial location microscopy data. A** Average scaled long-axis position predicted from MaxEnt models (solid lines) inferred from various Hi–C data sets (from[26]), including wild-type cells (black), rifampicin-treated cells (blue), and $\Delta smc$ cells (orange), together with results from microscopy experiments (adapted from[17]). Also shown are simulated data for a random polymer with *ori*-pole tether (dash-dotted gray line), and a simulated confined random polymer (dashed gray line), oriented such that *ori* is always on the left cell half. **B** The distribution of single-cell positions (scaled long-axis position) of chromosomal loci (blue: *ori*, red: *pilA*, green: *pleC*, orange: *podJ*), as predicted by the MaxEnt model (solid lines), together with previous experimental data from microscopy experiments (bars, adapted from[17]). To indicate experimental variability, the solid/transparent bars indicate the minimum/maximum measured by two different methods: FROS or FISH. To enable a direct comparison between model and experiment, the model values are distributed over the same number of bins as the experiment. The dotted lines indicate the distribution for a confined oriented random polymer as in **A**.

regions similar to those obtained in our super-resolution experiments. Importantly, the high-density regions in the modeled chromosome density plots correspond to underlying SuD structures (dashed lines in Fig. 4E). Thus, these results allow us to establish a connection between the SuDs predicted by our model and single-cell super-resolution data.

To investigate the influence of cellular processes on long-axis organization, we perform the two-point correlation and SuD structure analysis (Supplementary Note 9) on published Hi–C data of rifampicin-treated cells and a mutant lacking SMC ($\Delta smc$)[26] (Supplementary Note 13). Rifampicin treatment inhibits transcription, whereas deletion of SMC abolishes the loop-extrusion activity required to juxtapose the two chromosomal arms[53,56]. For both cases, our models predict an average localization along the long axis similar to those in wild-type cells

(Fig. 2A). However, the predicted long-axis correlations exhibit marked differences: for rifampicin-treated cells with inhibited transcription, anticorrelations between chromosomal arms are less pronounced (Fig. 3C upper left). In contrast, $\Delta smc$ cells display a broad regime with strong anticorrelations between loci on opposite arms (Fig. 3C lower right). These effects are reflected in the statistics of SuDs: upon inhibition of transcription, the SuDs contain 7% more genomic regions per domain than in the wild type. Despite this increased density, the transcription-inhibited cells show a similar overlap of SuDs (29% lower than for randomly paired arms). By contrast, $\Delta smc$ cells exhibit a similar average SuD density to the wild type (50 genomic regions per cluster on average), but a strong reduction of inter-arm domain overlap (48% lower than for randomly paired arms). Correspondingly, the anticorrelations between long-axis positions of chromosomal arms are much stronger for this mutant (Fig. 3C lower right). Thus, these results suggest that the action of SMC enhances interactions between SuDs, whereas transcription alters their density.

**Local chromosome extension coincides with high transcriptional activity, but only for one chromosomal arm**. The MaxEnt model provides access to local structural features that may be difficult to determine experimentally. Specifically, we consider the local chromosomal extension $\delta_i$, defined as the average spatial distance between two neighboring genomic regions of region $i$ (Supplementary Note 15). Interestingly, the $\delta_i$-profile exhibits an overall trend that is lowest at *ori* and *ter* (Fig. 5A), indicating that these regions are intrinsically more compact (Supplementary Note 15). In addition, pronounced peaks and valleys in the local extension are revealed at a smaller genomic scale similar to that of CIDs. The same structure appears for $\Delta smc$ cells, although their chromosome appears to be locally more compact than that of the wild type. By contrast, in rifampicin-treated cells, peak amplitudes are significantly suppressed, suggesting a link between local chromosome extension and transcription.

Previous work reported a connection between CID boundaries and highly transcribed genes[26]. Based on this observation and polymer simulations, it was suggested that high transcription creates plectoneme-free regions, physically separating CIDs. To further investigate the impact of gene expression activity on local structure, we compare the locations of local chromosome extension peaks in our MaxEnt model and the 2% most highly transcribed genes. Indeed, we observe a significantly increased overlap between the local chromosome extension peaks and the locations of highly transcribed genes, compared to a random distribution of peaks, but only for genes on the forward strand of the right *ori-ter* arm (0–2.0 Mb) (Supplementary Note 10). If the colocalization of local extension peaks by highly transcribed genes would only depend on the relative direction of transcription and replication, this should also occur for highly transcribed genes on backward strands on the left arm, which we do not observe. Thus, while our results indicate a connection between high local chromosome extension and the direction of replication and transcription of highly transcribed genes, the underlying molecular mechanism is still unclear.

**The chromosomal structure provides localization information in the cell**. The inferred structural features of the chromosome not only yield insights into the cellular organization, but they may also have functional significance: organizational features of the chromosome contain spatial information that could guide cellular processes. This spatial information depends on the degree of localization of genomic regions. Put simply, the localization information content of a genomic region increases with the

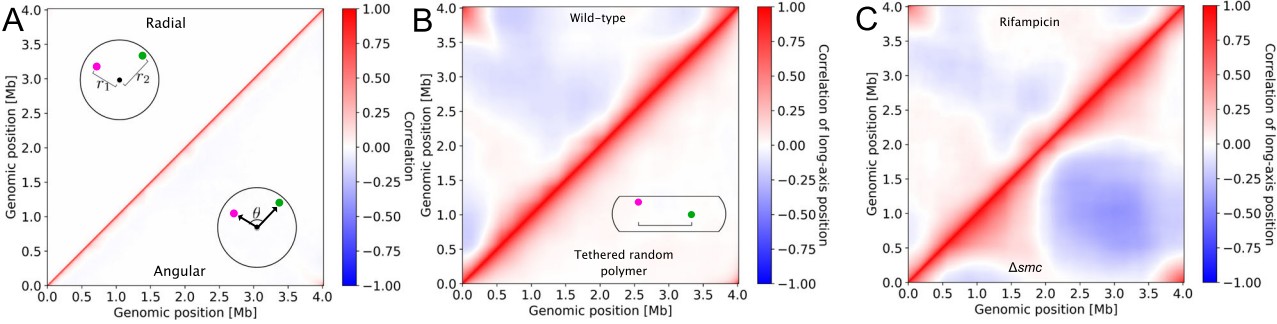

**Fig. 3 MaxEnt model predicts large-scale features of chromosome organization. A** Upper left corner: two-point correlations in the radial positions between genomic regions. Lower right corner: two-point correlations in angular orientations around the long axis. **B** Upper left corner: two-point correlations between long-axis positions for wild-type cells. Lower right corner: the same correlations for a model not constrained by Hi–C data, but with a tethered origin (tethered random polymer). **C** Correlations in the long-axis positions of genomic regions derived from Hi–C data[26] obtained for two modified conditions: cells treated with rifampicin to block transcription (upper left corner), and Δ*smc* cells (lower right corner).

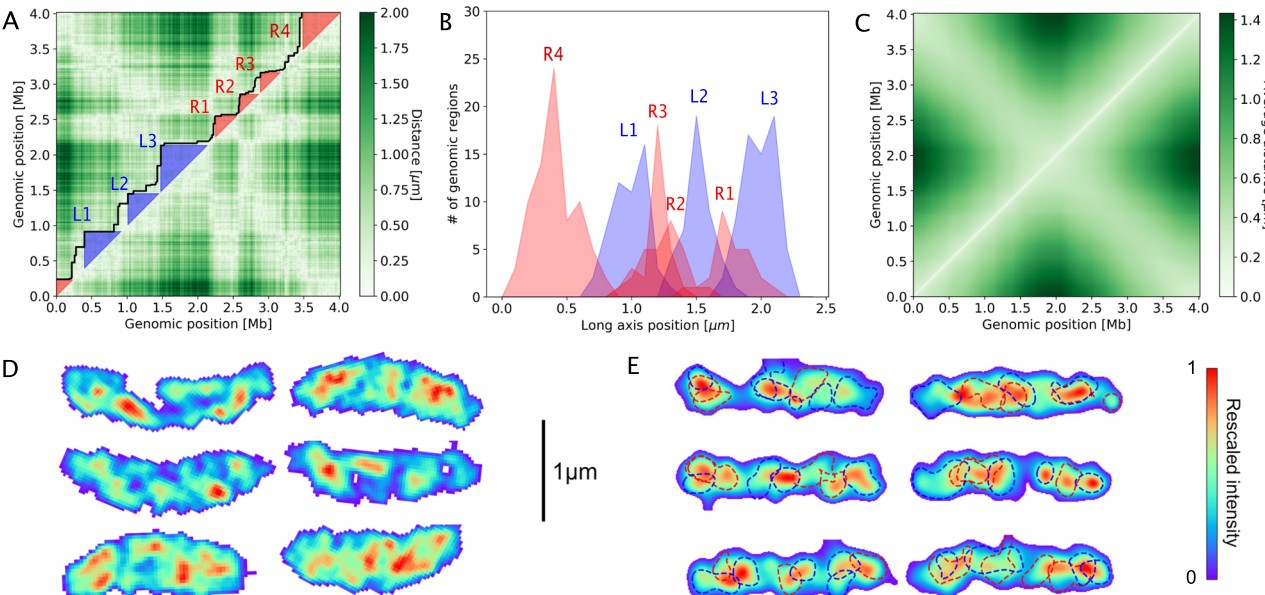

**Fig. 4 Long-axis organization is associated with Super Domain formation. A** Distance map for pairs of genomic regions for one chromosomal configuration. The inferred outlines (Supplementary Note 9) of Super Domains (SuDs) are indicated by a black line, with left/right-arm SuDs shaded blue/ red. **B** Long axis distribution of genomic regions in SuDs identified in the configuration depicted in A. **C** Average spatial distances between genomic regions. **D** Super-resolution microscopy images of DAPI-stained DNA inside six synchronized *C. crescentus* swarmer cells. The color code reflects the DAPI fluorescence signal at each pixel, rescaled so that the maximum is at 1 for each cell. **E** Chromosome density plot with the same scaling of several randomly chosen chromosome configurations from our MaxEnt model (with Gaussian blur applied that matches the experimental resolution). Dashed lines indicate the half-maximum density contour of each SuD (identified by the clustering analysis in Supplementary Note 9), with the line color indicating if a SuD predominantly forms on the right (0–2 Mb, blue) or left (2–4 Mb red) chromosomal arm.

precision of its cellular location, i.e., when the spatial distribution of the genomic region is more sharply peaked around a specific point in the cell. This localization information (introduced in the context of developmental patterning[57]) could for example be used to position proteins within the cell: a high relative affinity to a genomic region with high localization information increases the localization of this protein. This mechanism may be exploited to position protein droplets[58], through nucleation on specific chromosomal regions, e.g., droplet-like clusters of DNA-binding chromosome partitioning proteins of the ParB family[3].

Using our MaxEnt model, we can quantify how much localization information (Supplementary Note 14) is encoded by chromosome organization per genomic region (Fig. 5B). This chromosomal localization information is largest near *ori* and *ter*, providing 3 bits of localization information, equivalent to

reducing the localization uncertainty to one cellular octant. By contrast, a random polymer provides only 1 bit, enough to reduce localization uncertainty to one cell half. For comparison, with our coarse-grained description, maximal localization information of approximately 9 bits could be achieved. Thus, while this localization information metric indicates that the bacterial chromosome is substantially more ordered than a random polymer, it also highlights that the chromosome is far from having a rigid organization with a precise folded structure.

Comparing these results with those for modified conditions, we find that rifampicin treatment increases chromosomal localization information, whereas information is reduced in Δ*smc* cells, suggesting that SMC action and transcription have opposing effects on localization information. This localization information is just one example of how structural features in the organization

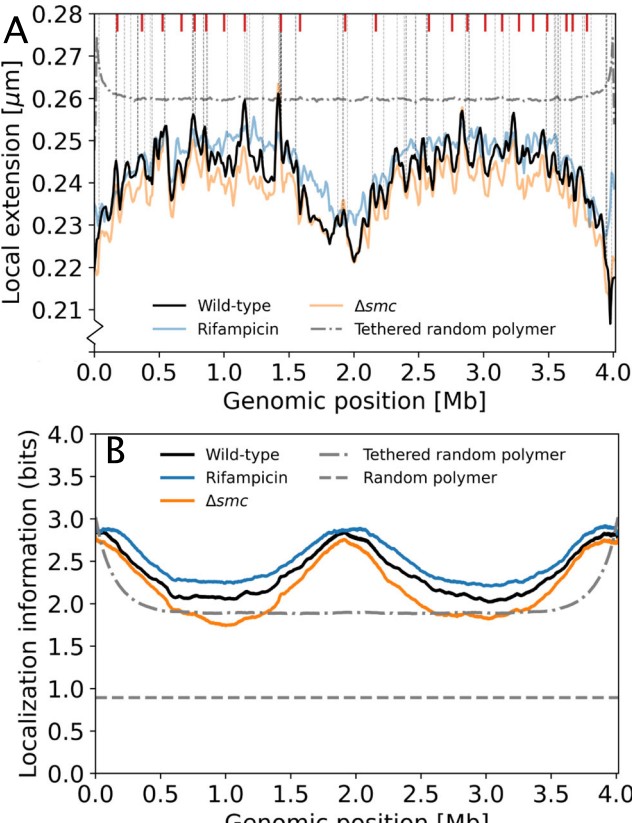

**Fig. 5 The MaxEnt model reveals local features and localization information encoded by chromosome organization. A** The local chromosome extension $\delta_i$ as a function of genomic position. $\delta_i$ is defined as the spatial distance between neighboring genomic regions of site $i$ averaged over all chromosome conformations. Model predictions are shown for wild-type cells (black), rifampicin-treated cells (blue), $\Delta smc$ cells (orange), and a pole-tethered random polymer (gray dash-dotted line). The locations of the top 2% highly transcribed genes are indicated by vertical gray dashed lines, the locations of CIDs determined in ref. [26] are indicated by red markers. **B** Localization information per genomic region in bits for wild-type (black), $\Delta smc$ (orange), rifampicin-treated cells (blue), a random pole-tethered polymer (dash-dotted line), and a random polymer (dashed line).

of the chromosome can be used to guide cellular processes. The MaxEnt approach provides a scheme to estimate the information available to the cell that is contained in the distribution of chromosome conformations.

## Discussion

We established a fully data-driven principled approach to infer the spatial organization of the bacterial chromosome at the single-cell level and applied this approach to normalized Hi–C data of the model organism *C. crescentus*. The predictive power of this MaxEnt model is confirmed by prior microscopy experiments[17] showing the distributions of axial positions of chromosomal loci within the cell. Contrary to previous modeling approaches, our MaxEnt model does not rely on an assumed connection between Hi–C scores and average spatial distances[21]. Instead, we can predict how these quantities are related: we recover the approximately linear relation between intra-arm genomic distance and spatial distance used as an input in refs. [21,33] (Supplementary Note 11). However, there are substantial region-to-region deviations in the resulting relation between Hi–C scores and average spatial distances, together with

significant correlations in distances between genomic regions. Previous approaches could not account for such deviations and correlations. This may explain differences in model predictions such as the helical chromosomal structure suggested in refs. [27,33], which we do not observe.

By design, the MaxEnt model yields the least-structured distribution of chromosome conformations consistent with experimental constraints, allowing us to investigate the degree of order in the bacterial chromosome. To do this, we considered two-point correlation functions in the cellular positions of genomic regions. We observe negligible correlations in the radial and angular coordinates, indicating an absence of organizational order in these directions. By contrast, there are pronounced long-ranged correlations along the long cell axis, indicating emergent order. This order is related to the observation of variable and delocalized clusters of genomic regions, which we term Super Domains (SuDs). These SuDs manifest in single-cell conformations and are consistent with high-density clusters observed in the *C. crescentus* chromosome by our super-resolution microscopy experiment (Fig. 3E). Similar blob-like structures have previously been observed with (super-resolution) microscopy for the chromosome of *Bacillus subtilis*[23] and *Escherichia coli*[13], suggesting that SuDs are also present in other bacteria. Our MaxEnt model indicates a spatial exclusion of opposing SuDs from different chromosomal arms, which we associate with the long-ranged anticorrelations in axial positions. The interplay between SMC complexes and transcription has been explored in prior work[28,59]. We find that transcription and SMC have opposing effects on SuD properties: inter-arm overlap between domains is reduced by transcription and increased by SMC, consistent with the idea that SMC links chromosomal arms[23,29,30,53].

At the smaller genomic scale of CIDs, we observe a characteristic pattern of local chromosomal extensions, being most compact at *ori* and *ter*. We speculate that the local compaction of the *ori* region may be due to the binding of nucleoid-associated proteins (NAPs)[1,2] such as the ParABS chromosome partitioning system[3,4]. The compaction of the *ter* region might be imposed by the recently discovered NAP ZapT[60], which specifically binds to this region of the chromosome, or by additional as-of-yet undiscovered NAPs. Interestingly, peaks in local extension tend to coincide with highly transcribed genes, but only for the forward strand of the right chromosomal arm (Supplementary Note 10).

From our MaxEnt model, we obtain an estimate of the chromosomal localization information per genomic region. This information reaches up to 3 bits around *ori* and *ter*, equivalent to a localization uncertainty in the cell of one cellular octant. We speculate that such localization information encoded by the organization of the chromosome could be exploited for sub-cellular positioning of proteins and protein droplets[58] or for the regulation of transcription of genes, as was observed in[20].

Our approach resides in the class of static Maximum Entropy approaches, which make no assumptions or predictions about the underlying dynamics, as opposed to dynamical maximum entropy models or maximum caliber models (see for instance[61,62]). Further model limitations are set by the available input data: organizational features that cannot be faithfully encoded in population-averaged Hi–C data might be absent in the MaxEnt model. The resolution of Hi–C data is limited to 10 kb for the data sets analyzed here, implying that any organizational features below this genomic length scale cannot be explored with our model. However, our approach is not limited to interpreting Hi–C data and can be extended towards an integrated MaxEnt model, simultaneously constrained by both Hi–C and microscopy data (Supplementary Note 17). Furthermore, our approach may be generalized to other prokaryotes, including

systems with replicating chromosomes and multiple replicons, as well as eukaryotes, paving the road for unraveling all information on chromosome conformations at multiple length scales, elucidating single-cell variability and population averages.

## Methods

Here, we consider Hi–C data (replicate 1 of the BglII Hi–C data) on *C. crescentus* newborn swarmer cells published in ref. [26], which have a single, non-replicating chromosome. However, due to imperfect synchronization, a small fraction of cells are included in these experiments in which processes such as chromosome replication and segregation have initiated, which will be reflected in the Hi–C map[27,33]. Before inferring a MaxEnt model, we apply a data-processing scheme to filter out contributions from cells with replicating chromosomes (See Supplementary Notes 5–6). However, we also provide a MaxEnt model inferred directly from the unprocessed Hi–C data (See Supplementary Note 7) and MaxEnt models inferred from Hi–C data sets for replication-arrested cells[25] (See Supplementary Note 8). While there are small differences between the different models, the central behaviors from the MaxEnt model reported in the main text are similar in all cases.

Our algorithm (Supplementary Notes 3,4) requires two length scales: the dimensions of the cellular confinement and the lattice spacing. As cellular confinement, we use a cylinder capped with hemispheres with the dimensions of a newborn swarmer cell minus the cell envelope: $0.63\,\mu m \times 2.2\,\mu m$ (Supplementary Notes 1–2), which is assumed to be the same for all cells. A more detailed representation of the cellular confinement shape does not appear to affect our main results (Supplementary Note 17). To set the coarse-graining scale of our MaxEnt model, we experimentally determined the distribution of spatial distances between subsequent Hi–C bins. Specifically, the lattice spacing, $b$, is set by the average spatial distance between consecutive 10 kb regions (the Hi–C bin size). To determine this parameter, we probed the physical distance of two loci separated by 10 kb in five different regions of the chromosome, using an approach comparable to[63,64]. To this end, we constructed strains whose chromosomes contained two independent arrays of transcription factor binding sites (comprising 10 LacI or TetR binding sites, respectively) inserted at the proper distance (Supplementary Note 1). The sub-cellular positions of these arrays were then determined by producing the respective fluorescently labeled transcription factors (LacI-eCFP and TetR-eYFP) at very low levels, based solely on the basal activity of the inducible promoter driving their expression. Swarmer (G1-phase) cells were imaged immediately after isolation, and the localization of the two arrays was determined with sub-pixel precision by fitting a 2D Gaussian to the acquired images. The Euclidean distances between the two arrays were calculated, taking into account correction factors for a systematic shift produced by the set-up (see Methods for further details) and are shown in (Table S5). The average distance between genomic loci 10 kb apart were found to be $129 \pm 7$ nm, implying a lattice spacing $b = 88$ nm (Supplementary Note 2). For the selection of cells in Fig. 4D, cells with approximately the average newborn cell length ($2.3 \pm 0.2\,\mu m$ (Supplementary Note 2.2)) were chosen. For each cell, out of the z-stack, the plane that corresponded to the mid-cell being in focus was selected. For the calculation of single-cell chromosomal density plots (Fig. 4E), a Gaussian blur was applied, whereby the resolution in the z-direction (300 nm) and in the x and y directions (120 nm) were set to match the experimental resolution.

**Reporting summary**. Further information on research design is available in the Nature Research Reporting Summary linked to this article.

## Data availability

Data supporting the findings of this manuscript are available from the corresponding author upon reasonable request. A reporting summary for this article is available as a Supplementary Information file. A sample of chromosome configurations generated by the MaxEnt model is available on GitHub[65].

## Code availability

The code generating the data and implementing the analysis presented in the manuscript is available on GitHub[65].

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

## Acknowledgements

We thank Tung Le for helpful discussions and for generously making experimental data available. In addition, we thank Ben Machta for inspiring discussions, Karsten Miermans and Lucas Tröger for valuable input for the simulations, Gabriele Malengo (Facility for Flow Cytometry and Imaging, MPI Marburg) for help with the super-resolution microscopy, and Maritha Lippmann for excellent technical assistance. This research was funded by the Deutsche Forschungsgemeinschaft (DFG, German Research Foundation, Project 269423233-TRR 174). J.M. is supported by a DFG fellowship within the Graduate School of Quantitative Biosciences Munich (QBM).

## Author contributions

C.P.B. conceived the project; J.J.B.M. performed analyses, simulations, and analysis of microscopy data, J.J. and J.J.B.M. developed the MC algorithm, M.T. and M.C.F.vT. conceived microscopy experiments, M.C.F.vT. performed microscopy experiments and analyzed microscopy data, C.P.B. and J.J.B.M. wrote the paper with input from all authors.

## Funding

## Competing interests

The authors declare no competing interests.
