## [Peer Review File · Nature Communications]

Reviewer #1 (Remarks to the Author):

**** Summary

The authors build a maxent model of circular bacterial chromosome conformations, where the chromosome is represented as a polymer on a discrete 3D lattice and maxent constraints are proportional to the observed high-C contact probabilities. This provides a full probabilistic description of conformations allowing non-trivial predictions to be made about emerging order. The model provides localization predictions validated against microscopy that are in agreement with data which is inconsistent with random polymer models. Maxent suggests global order in the form of SUDs, whose signature are large-scale long-axis correlations in position of different genomic markers. This is worked out for two extra biological perturbations; the authors introduce the concept of localization information, similar to recently introduced formalization of positional information in development.

**** Recommendation

This is a beautiful paper and I recommend publication after addressing several points to improve its clarity and breadth.

**** Major

1) How do you know there is a causal link: "We find that these intra-arm anticorrelations result from the spatial exclusion of large genomic clusters between the two chromosomal arms"?

2) Is there any extra correlation structure GIVEN SUDs? Right now, it appears that correlations are washed out if you average over the whole ensemble (3F) even though conditional on the configuration there is large-scale structure (3D). Does it make sense to look at radial / angular correlations within / across SUDs? In general, I missed a bit more details about what kind of structures these SUDs are: e.g., what is the histogram of their numbers in the ensemble on both arms, are there sections of the chromosome that are ever not in a SUD, how diverse are SUD configurations (that is, if you make a long sampling and construct equivalents of 3E, what's the diversity of SUDs that you see -- do a small number of them repeat)? Is there any way to figure out which energy couplings are necessary and sufficient for the same type of SUD global order (e.g., can you parametrize the epsilon matrix in a simple way to recover the same qualitative order)?

3) A more risky, but potentially interesting exploration of your model: if you sample from a Hamiltonian Eq (5) with properly reconstructed epsilon, but create an ensemble at lower temperature (I know this is not equilibrium, but you can treat T just as a free parameter that increases / decreases interaction strengths), do you see at low T more clear order emerging, in particular, are SUDs clearly connected to the minima of the energy function? Perhaps one can derive interesting hypotheses from this construction, as was done in neurons (Tkacik et al, PLOS CB 10: e1003408 (2014); Berry & Tkacik, Frontiers Comput Neurosci 14: 20 (2020)).

**** Minor

1) "Our MaxEnt model does not rely on equilibrium assumptions,"

In what way do the other approaches? A bit more of discussion on this important point would be useful.

2) Page 2: Unclear "a polymer on a 3D cubic lattice, with a subset of monomers representing N

genomic regions. " What is "genomic region" here? What is the dimension of the lattice relative to the mentioned 10^5 constraints? What is the dimensionality of the sigma? In general, I got stuck for a while trying to make it absolutely clear to myself how the authors really represent their circular polymer (since there are many ways to represent its configurations), and being very slow and pedestrian here in the setup would help. Maybe a small schematic with a 3D lattice and a polymer inside; also mention that by construction of your ensemble and its moves you will ensure the polymer is always connected and circular (else one could incorporate these features as hard maxent constraints with infinite coupling in H, which I initially looked for but didn't find). In short, explain more clearly how the polymer is represented.

3) Broken reference after equation S4 in the SI.

4) Where does Eq 6 comes from -- what is the motivation for removing degeneracy due to unknown c by Eq 6? SI S3.2 does not explain, since it starts by assuming Eq 6 to derive what the energy shift should be.

5) Do you need to imply any regularization for epsilons? How come that pairs for which $f_{ij} = 0$ are not assigned epsilon -> infity? I could only see this happening either because (i) the algorithm does not fully converge in epsilon space (although it may have converged until tolerance in the constraint space) OR, more interestingly, if (ii) the polymer nature of the problem (= loop) regularizes epsilons "automatically"...

6) You may consider citing De Martino et al, "Statistical mechanics for metabolic networks..." Nat Comms 9: 2988 (2018), along with other examples of maxent applications to biology. As in your case, there too sampling needs to explore a constrained space (as in your case, for the circular connected polymer), so special moves in the flux space are devised to be consistent with constraints.

Gasper Tkacik

Reviewer #2 (Remarks to the Author):

In "Learning the distribution of single-cell chromosome conformations in bacteria reveals emergent order across genomic scales", Messelink and co-workers develop a maximum entropy approach to reveal aspects of spatial chromosome organization from Hi-C data as input. First, microscopy experiments are used to calibrate coarse-grained polymer simulations at short (<10 kb) length-scales. Then, using their iterative model optimization approach (MaxEnt), the authors infer interaction coefficients (ϵ_{ij}) between chromosomal loci which defines the maximum entropy configurational ensemble of chromosomal states. This ensemble of chromosome states is used to make testable predictions about the spatial positioning of loci within the bacterial nucleoid of *Caulobacter crescentus*. The approach is validated by comparing their inferred 3D chromosome positions to previously published FROS and FISH imaging data, showing good agreement. Among the interesting new findings, the authors find that local chromosome extensions and exclusions are present in the 3D spatial positioning inferred from their model; these are nice hypotheses that can be tested experimentally in the future.

The authors make an important contribution to the field of chromosome biology. This work will both be of interest to data scientists and theorists studying chromosome structure, and to a wider audience of biologists, as the authors generate testable hypotheses for the spatial positioning of

specific chromosomal loci. Importantly, the method developed here, while demonstrated for *Caulobacter crescentus*, is readily generalizable to eukaryotic genomes as well, and may open new avenues for generating testable hypotheses in other species of bacteria with potentially different nucleoid geometries and chromosome organizations.

Overall, the method presented herein represents one of the most sensible (non-hypothesis driven) treatments of the “inverse problem” of chromosome structure elucidation from Hi-C data. The approach relies minimally on assumed distributions of spatial distances (except for the setting the coarse-graining length scale, which can be otherwise inferred), or specific detailed knowledge of the positioning of chromosomal loci as inputs. Moreover, this approach does not provide a ‘single representative chromosome structure’ as in some methods in the literature; they further introduce the notion of localization information which helps to formalize our understanding of chromosome positioning uncertainty and may help towards dispelling a prevalent conception of the ‘chromosome structure’ as static, or highly ordered structures. All combined, these features make the approach proposed here by Messelink et al. potentially quite valuable to the scientific community at large. I recommend this article for continued consideration at Nature Communications. However, I recommend several important revisions to augment the rigor and clarity of the study before publication. Outlined below are my concerns, as well as some minor comments.

Major comments

1. The article main text is well written and generally clear. However, it is largely missing a discussion of the limitations of the MaxEnt method. It seems that some of the most important assumptions of the method are largely unstated, and readers are left to figure out what these are. Some suggestions of things to discuss include the assumption of no excluded volume on the polymer chain [see point 6]. Please also elaborate on how multiple-fork replication, multiplicity of chromosomes, and poorly mapping regions in Hi-C will affect your results. For transparency and as a service to the readers, I strongly recommend that the authors highlight these (and other) limitations to their method in the main text and discuss their implications.

2. Reproducibility, code and data availability: the proposed MaxEnt method is overall pretty complicated to a non-expert as it involves 3D polymer simulations for the forward algorithm, and an iterative updating scheme for the inverse algorithm. I strongly recommend the authors make (a) the source code available, with some examples on how to run it, and (b) make data available for their MaxEnt generated conformations. The former will be important to minimize the barrier to entry for others looking to employ these methods, and the latter will be important for the wider community (including this reviewer) to more carefully evaluate the results of the author’s modelling on their own.

3. Data treatment: raw Hi-C data is “cleaned up” (as described in Supplemental Section S5) to remove what the authors deem to be an “artefact” in the data (i.e. increased frequency of ori-ter contacts); however, the cleaning up procedure does not appear statistically or physically valid, and also relies on a number of subjective choices. I do not believe this treatment of the Hi-C data is acceptable for the following reasons:

a) The so-called artefact that is corrected for/removed (i.e. a higher frequency of ori-ter contacts than other similarly separated loci) is not really an artefact. These data points likely represent a real biological phenomenon arising from the active motion of the origin towards the terminus, and thus are an integral part of the biological ensemble of chromosomal states for that population of cells. The removal of interaction pairs below a certain frequency hinges on an interpretation (not solid

data) of what is going on (i.e. it is indirectly inferred by Hi-C and microscopy) and does not seem scientifically sound.

b) On a more subjective note, this filtering is motivated using “outside knowledge” about the system, which is contrary to the spirit in which the authors frame their method (i.e. the claim that the inference is entirely Hi-C data-driven).

c) Most importantly, I wonder if the proposed data curation may have arisen from a misunderstanding of contact probability decay curves: in Region II where the data envelope is “fit” and points are removed does not appear to be noise. Instead, the “upwards trend” of the contacts beyond 2 Mb naturally arises when the authors perform log-binning of genomic coordinates for a circular chromosome. This same “upwards trend” is present even for the most naïve models of chromosome conformations (e.g. a purely Gaussian circular chain with no confinement) and it does not constitute “spurious noise” as the authors claim. I believe it is thus incorrect for the authors to fit these interactions to a power-law, calculate two standard deviations, and subtract the resulting interaction frequencies from the data.

To remedy these issues, I strongly recommend that the authors repeat the analyses without removing these contacts, or otherwise demonstrate more rigorously how their proposed treatment is statistically and physically sensible.

4. In the section, “MaxEnt Model of the *C. crescentus* Chromosome Quantitatively Captures Measured Cellular Localization”, the authors state that, “[They] orient cells by setting the ori pole in the cell-half containing ori”. In other words, from the inferred chromosome conformations derived by the MaxEnt method, the ori’s are consistently placed in the same cell half. Unfortunately, the authors seemingly do not apply this procedure to the polymer simulations that they use as benchmarks (i.e. specifically, the “random polymer” simulations), which gives rise to misleading results. To illustrate this point, in Fig 2 (top panel): the “random polymer” simulation average long-axis position is a constant 0.5 over the entire chromosome length. I would firstly not expect a flat distribution of points for the average long-axis positioning if the “orienting procedure” were systematically applied (e.g. for the ori positioning, I would expect to see it closer to ~0.25 (and not at 0.5)); this suggests that the authors are treating the MaxEnt conformations and other polymer simulation conformations differently, and gives a false impression for the relative goodness of their approach. I suggest that the authors revisit these comparisons and fix the analyses appropriately.

5. Relatedly to point 4, the authors use two types of polymer simulations as benchmarks by which to compare the MaxEnt method. The comparisons are to the “tethered random polymer” & “random polymer” throughout the text. However, a more appropriate model is the “tethered random polymer with juxtaposed chromosome arms”, or a simpler “polymer with juxtaposed arms but no tethering” model. This comparison also seems to be a natural extension of their previous work, where they have developed such simulation models (e.g. as in Ref. 27, for the case of active SMCs and tethered origins); moreover, the models I suggest above are more representative of the collective body of knowledge, as it incorporates information that has been in the field for >5 years (i.e. the fact that chromosome arms are juxtaposed, and tethered to the ori). I recommend these additional comparisons to be made, at the very least in the supplementary materials.

6. In the analysis of Super Domains the authors use the language “excluded” where it perhaps could lead to misunderstanding: e.g. “We find that these intra-arm anticorrelations result from the spatial exclusion of large genomic clusters between the two chromosome arms.” Based on the unstated model assumption of no excluded volume interactions (from their lattice polymer simulations), no two loci can be truly “excluded” from one another. Exclusion may only occur in the statistical sense: I

wonder if the authors can demonstrate that such exclusions arise in single cell conformations, or if these are just statistical averages present from the population of cells. The notion of “spatial exclusions” is also seemingly at odds with the alignment of the left and right replisomes as seen by Hi-C. I suggest the authors address whether these exclusions are present within individual chromosome conformations, and address how said exclusions are also consistent with the 3C contacts, and consider revising their language in the main text.

7. In Figure 2B, the single cell distributions of chromosomal loci by FROS do not seemingly match with the values plotted in Figure 2A. For instance, a locus tagged at the ori seems to have a scaled mean long-axis position of ~ 0.1 in Figure 2B but averages out to a position of ~ 0.2 in Figure 2A. Can the authors comment on the difference?

Along these lines, can the authors comment on how their MaxEnt model (and well as the tethered random polymer model) finds the “exact cell positioning” of the ori to be at 0.2 along the long axis? It seems rather non-trivial that this should be the default placement... is it externally imposed?

8. Supplemental section S2.2 is confusing and seemingly inconsistent with S3.1— more details are required. Specifically, it states “In this representation [of the polymer], the position of each even monomer indicates the cubic unit cell occupied by the center of a Hi-C chromosomal region”. This is seemingly in contrast with section S3.1, which states that “Each 4th monomer represents the location of the center of a genomic region, with three monomers in between to ensure Gaussian statistics between subsequent centers of genomic regions”. I was also wondering if the authors could elaborate on how their coarse graining leads to 7 possible distances (length $0b$, $\sqrt{2}b$... , $4b$); perhaps some illustrations/sketches may help the reader visualize where these are coming from.

9. The authors develop a concept of localization information, which uses the ensemble of 3D polymer conformations to define the likelihood of being able to find a particular chromosomal segment in a pre-defined nucleoid compartment. It took me a while to appreciate the significance of this result. I think this is an important concept for the field moving forward, but it is also very surprising that the localization information is so low, with the highest localization information value of 3 (or the equivalent of localizing to 1/8th of the cell). This highlights the inherent stochasticity of 3D chromosome structures. I think the paper could benefit from a slightly more elaborate discussion of this subtle concept; perhaps the authors could give more examples to build reader intuition for the localization information (and its meaning) and to contextualize its biological significance.

Minor comments

1. The authors claim their method is compatible with non-equilibrium dynamics, but do not elaborate on how. Please clarify this statement, since: 1) it appears that by defining the ϵ_{ij} as a temporally invariant quantity, or 2) in the absence of any limit cycles, or 3) active forces in the model, the proposed method is not really addressing the question of non-equilibrium dynamics...

2. In page 2: “To this end, we introduce constraints to the entropy functional: [Eq. 2].” Strictly speaking the equation is not for the constraints themselves but for the Lagrangian... a reference to the method would be helpful, and a tweak in wording. E.g. “To this end, we introduce constraints to the entropy functional by introducing the Lagrangian function: [Eq. 2].”

3. A reference is needed for the mapping of Eq. 4 to the confined lattice polymer, Eq. 5.

4. The supplement has a bibliography but in-text references are missing to these cited papers.
5. In section S2.1 what is the rough value for the “drift between frames” before the correction?
6. S2.1 is written in a slightly complicated and confusing manner. The descriptions can be simplified (e.g.). Moreover, it stated that the loci distances are Gaussian (page 8) referring to Figure S1. The distances are not Gaussian as they only have positive values...
7. More details are needed for the Forward simulation (Fig. S3.1): what are the initial conditions for the simulations? Are configurations equilibrated?
8. Some equations are missing, likely in the LaTeX file compilation (supplement page 12, after Eq. S4).
9. Sometimes it was a bit confusing what is being referred to (i.e. whether S12 is equation S12 or section S12 – e.g. page 9 of supplement). Please consider prefacing all the equations in with Eq. SX, or sections with “Section SX”. Note: section S9 also has references to Figs. SS12, and SS13 (extra S) as does Section S12 (SS18, SS19).

Reviewer #3 (Remarks to the Author):

The manuscript describes a Maximum Entropy approach to reconstruct single-cell 3D structures of bacterial chromosome based on population-based Hi-C maps. The modelling part is done with a lot of rigor, and likely calculates the maximum entropy ensemble correctly.

The application of the method to the bacterial system is of potential interest; however, it is complicated by the fact that there are no known ways in which organization of bacterial chromosomes influences important cellular processes (e.g. gene expression). This makes it difficult to evaluate and interpret results of the manuscript in a broader context. The presented model does not reveal any new mechanisms of chromosomal folding, or any influences of chromosomal folding on cellular processes.

Analyses of the model are not done very carefully. The authors “brush off” the fact that they see good agreement on one chromosomal arm and not on another. They do not provide any justification of the SuD domains or any single-cell data that would support them.

In general, I would much rather see applications of this method to mammalian systems. It would be interesting to see this model applied to other Hi-C maps, such as TADs in mammals, micro-C data with grids of loops (Krietenstein et al, 2019), etc. I strongly encourage the authors to consider these systems in this or subsequent manuscripts.

I believe that the model presented by the authors correctly infers a maximum entropy ensemble subject to the constraint of a Hi-C map. However, the model is only applied to the bacterial data and does not make any conclusions relevant to biology. Moreover, I have doubts about applicability of maximum entropy reconstructions to Hi-C data in general. Therefore, I believe that the manuscript is of limited interest to the broad scientific audience, and is better suited for a more specialized journal.

Major comments

1. The result that areas of large extension correspond to active genes on one chromosomal arm is perplexing, and warrants much deeper investigation. However, the manuscript just “brushes it off” by suggesting some kind of symmetry breaking: “Thus a feature is required to break this symmetry.”.

I strongly disagree with this approach. I think the authors have to get down to the root of the difference between the two arms. The agreement of the peaks of local extension with the active genes seems rather good on the first arm; they may agree for the first couple peaks on the second arm, but then the agreement disappears.

There are few possible explanations. One of them is that highly expressed genes on the second arm do not produce a strong signature in the Hi-C maps (i.e. do not correspond to the peaks of insulation). Perhaps, the authors should compare insulation in Hi-C maps with local extension, and then insulation with highly expressed genes, to see which of the two relationships are broken. If the former, then it suggests a potential problem with the algorithm. Another explanation is that the relationship between local extension and insulation in Hi-C maps is non-trivial (e.g. having lots of insulation peaks on an arm does not produce much local extension). This could be studied by feeding synthetic Hi-C maps into the model. Finally, there could be a bug in the code that leads to gradual de-synchronization of the two plots.

2. The anti-correlation between the two chromosomal arms is an interesting result. I believe that it has a plausible biological explanation, however, a mechanistic model such as the one in (Le, 2013) would be needed to test mechanistic hypotheses. Normally, tethering ori at one end of the nucleolus aligns the two arms together. However, if the ori were to be attached not strictly at the pole of the nucleolus, one of the arms would fill the ori end of the nucleolus, and then proceed towards the terminus side of the nucleolus. Another arm would go towards the terminus right away due to steric exclusion. The whole chromosome would “rotate” with (ori – epsilon) or (ori + epsilon) being where ori should be theoretically. This would lead to anti-correlation between positions of the two arms.

It would not arise in a long polymer considered as a control because it is too long. However, I believe that the effect would arise naturally if the authors were to compute similar correlations in a circular bottlebrush model as in (Le 2013). A mechanistic explanation of why this anticorrelation arises, and what are the biological implications of it, would be very helpful for understanding the results of the model.

3. The authors claim to have discovered a new kind of domains in single cells, termed SuDs. Those domains appear only as a product of the model the authors present. The authors do not establish any connection between SuDs and other biological data, and do not verify SuDs by any single-cell experiment.

The bioinformatics analysis of SuDs is equally lacking. Are SuD boundaries enriched in something? Can you predict them from biological data? What are average positions of boundaries? Can you see them in microscopy?

Even if these statistics are presented, I would still take SuDs with a grain of salt, because the authors have no direct single-cell measurements of the domains that they discover. Other single-cell domains were met with a degree of controversy in the field despite being observed directly in single-cell experiments (“Contact domains” in Flyamer et al, 2017; and “TAD-like structures” in Bintu et al, 2018 come to mind).

4. On a broader scale, I have doubts about the validity of maximum entropy methods to infer single-cell conformations. In actual cells, folding of chromosomes is a consequence of a fairly complicated and potentially active process involving many architectural proteins. How can we be sure that the conformational ensemble achieved by this process matches the maximum entropy ensemble?

Imagine that all you have is an average 2D projection of a building (or many buildings). With a maximum entropy approach, a reconstruction of the ensemble of buildings, subject to the constraint of this projection, would uniformly fill the space between the ground and the roof with uniformly-distributed building materials: windows/walls on the outside, walls/doors in the center. In reality, a building was obtained through a particular process of making it, that put building materials into walls and floor, and left rooms empty. This reduced the entropy, which is expected for any building process. Same logic could be applied to the enzymes that fold chromosomes: they don't have to follow the maximum entropy ensemble; they likely reduce entropy and achieve only a particular subset of this ensemble. The MaxEnt method derives a most general potentially non-equilibrium set of conformations matching the constraints, while the reality is likely a not-so-general non-equilibrium set of conformations matching the constraints.

If one were to apply MaxEnt approach to maps of mitotic or prophase chromosomes, would they recover the nested condensin I/II loops with condensins forming a scaffold in the center? My guess would be that they won't: the ensemble created by making the loops-on-the-scaffold structure is too constrained, and occupies a tiny sub-space in the whole maximum entropy ensemble consistent with fairly featureless mitotic Hi-C maps.

REVIEWER COMMENTS

Reviewer #1 (Remarks to the Author):

**** Summary

The authors build a maxent model of circular bacterial chromosome conformations, where the chromosome is represented as a polymer on a discrete 3D lattice and maxent constraints are proportional to the observed high-C contact probabilities. This provides a full probabilistic description of conformations allowing non-trivial predictions to be made about emerging order. The model provides localization predictions validated against microscopy that are in agreement with data which is inconsistent with random polymer models. Maxent suggests global order in the form of SuDs, whose signature are large-scale long-axis correlations in position of different genomic markers. This is worked out for two extra biological perturbations; the authors introduce the concept of localization information, similar to recently introduced formalization of positional information in development.

**** Recommendation

This is a beautiful paper and I recommend publication after addressing several points to improve its clarity and breadth.

We thank the reviewer for these supportive comments.

**** Major

1.1) How do you know there is a causal link: "We find that these intra-arm anticorrelations result from the spatial exclusion of large genomic clusters between the two chromosomal arms"?

The MaxEnt model predicts both the long-ranged negative intra-arm correlations and SuDs, and we provide evidence for a relation between these two observations. Specifically, we find that the SuDs positioned on opposite sections of the two chromosomal arms exhibit a correlated cellular localization pattern. To better illustrate this, in addition to new Fig. 5 (Fig. 4 in our original submission), we now provide computed density plots of chromosomes based on our MaxEnt model. These simulated density plots represent predicted microscopy images of single chromosomes in a living cell (new Figure Fig. 4E). In the computed density plots, the SuDs show up as clearly defined high-density regions, which we compare to new super-resolution experiments of single cell chromosomes that indeed exhibit similar high-density clusters (new Figure Fig. 4D). The computed density plots illustrate that SuDs from different arms tend to spatially exclude each other, which is also borne out of our more detailed correlation analysis discussed in section S9.1. As a result of this tendency to spatially exclude, chromosomal regions belonging to SuDs on opposing sections of the two arms, are expected to fluctuate in an anti-correlated fashion. The connection between the negative intra-arm long-axis correlations and the spatial exclusions of SuDs is further supported by the results on the Δsmc mutant. For Δsmc cells, we find that both the negative intra-arm correlations are stronger (Fig. 3c) and the spatial exclusion between SuDs on opposing sections of the two arms is more pronounced (Main text section 'Large-scale chromosome organization primarily characterized by long-axis correlations associated with Super Domains'). Based on this evidence, we argue that the spatial exclusion of large genomic clusters between the two chromosomal arms is *associated* with the intra-arm anticorrelations. While we believe that a causal link is plausible, we agree that it is difficult to make strong claim about causal connections.

To make our formulation of this point more precise, we also updated the main text in the following way:

Page 4, lines 287-289:

"We find that these intra-arm anticorrelations are associated with large genomic clusters, which we term Super Domains (SuDs)."

Page 4, lines 307-310:

"This exclusion behavior is expected to generate negative intra-arm anticorrelations for pairs of genomic regions with similar average axial positions."

Page 5, lines 361-363:

"Correspondingly, the anticorrelations between long-axis positions of chromosomal arms are much stronger for this mutant (Fig. 3C lower right)."

In the Discussion, lines 500-511:

"Our MaxEnt model indicates a spatial exclusion of opposing SuDs from different chromosomal arms, which we associate with the long-ranged anticorrelations in axial positions."

We included new predicted and measured single cell chromosome density images, which we connected to the SuD analysis (Fig. 4D,E)

1.2) Is there any extra correlation structure GIVEN SUDs? Right now, it appears that correlations are washed out if you average over the whole ensemble (3F) even though conditional on the configuration there is large-scale structure (3D). Does it make sense to look at radial / angular correlations within / across SUDs? In general, I missed a bit more details about what kind of structures these SUDs are: e.g., what is the histogram of their numbers in the ensemble on both arms, are there sections of the chromosome that are ever not in a SUD, how diverse are SUD configurations (that is, if you make a long sampling and construct equivalents of 3E, what's the diversity of SUDs that you see -- do a small number of them repeat)? Is there any way to figure out which energy couplings are necessary and sufficient for the same type of SUD global order (e.g., can you parametrize the epsilon matrix in a simple way to recover the same qualitative order)?

Thanks for all these insightful suggestions. Within a SuD, we find that chromosome organization shows similarities to a random confined polymer in several respects. The angular correlations within a SuD decay much more rapidly than for a random position along the chromosome (Figure R1A&B), and the mean distances for genomic regions within a cluster show similarity to those corresponding to uniform, independent distributions of genomic regions (Figure R1D). However, we do find some additional structure if we consider the radial correlations within a SuD: we observe anticorrelations between the ingoing and outgoing segments from the cluster center (Figure R1C).

Considering the locations and sizes of SuDs, we find that there are on average ~4 SuDs on each chromosomal arm (Figure R1E), and the average SuD size is highest around the origin and terminus (Figure R1F). Additionally, we find that on average, a fraction of 0.77 genomic regions is part of a SuD.

We included panels E, F and G in a new supplementary section SI S9.2.

Figure R1 A) Upper left triangle: correlations in the angles θ between regions with a SuD, with θ measuring the angle between the short-axis coordinates relative to the cluster center. Lower right triangle: same correlation for a randomly chosen site on the chromosome. B) Upper left triangle: correlations in the angles ϕ between regions with a SuD, with ϕ measuring the angle between a short-axis and a long-axis coordinate relative to the cluster center. Lower right triangle: same correlation for a randomly chosen site on the chromosome. C) Upper left triangle: radial correlations between regions with a SuD relative to the cluster center. Lower right triangle: same correlation for a randomly chosen site on the chromosome. D) Upper left triangle: average distances between regions within a SuD. Lower right triangle: average distances for an ideal gas contained within a SuD radius. Analyses A-D are performed on SuDs containing at least 30 genomic regions. E) Distribution of the number of SuDs across configurations for the left arm (blue) and the right arm (orange). F) Average size of the SuD as a function of genomic position, given that the corresponding genomic region is part of a SuD. G) Probability of a cluster center being within 50 kb of a genomic region, as a function of genomic position.

Additionally, we made the following additions in the main text on page 4, lines 295-298:

"On average, 77% of genomic regions is part of a SuD, each chromosomal arm contains ~4 SuDs, and each SuD contains ~50 genomic regions (SI Figure S21)."

There are several other interesting features of SuD organization that could be explored, such as the correlations in positions of SuDs, connections between SuD statistics and locations of Contact Interaction Domains, the diversity of SuDs, and necessary features of the Hi-C map to generate SuDs. We feel these questions are very interesting and deserve a thorough investigation, and we would choose to give these analyses the needed space in future work. We are currently developing further experiments on *C. crescentus* to explore SuD properties *in vivo*, and plan to combine the results with deeper theoretical investigations of SuDs in the MaxEnt model. We thank the reviewer for raising several interesting questions for further investigation.

1.3) A more risky, but potentially interesting exploration of your model: if you sample from a Hamiltonian Eq (5) with properly reconstructed epsilon, but create an ensemble at lower temperature (I know this is not equilibrium, but you can treat T just as a free parameter that increases / decreases interaction strengths), do you see at low T more clear order emerging, in particular, are SUDs clearly connected to the minima of the energy function? Perhaps one can derive interesting hypotheses from this construction, as was done in neurons (Tkacik et al, PLOS CB 10: e1003408 (2014); Berry & Tkacik, Frontiers Comput Neurosci 14: 20 (2020)).

Thanks for this insightful suggestion. We agree that this is a very interesting direction to explore, especially in connection with investigating necessary criteria for SuD formation. We feel that such an analysis would fit well with the further investigation of SuD structure discussed in point 1.2, and are in the process of addressing this question in future work.

**** Minor

1.4) "Our MaxEnt model does not rely on equilibrium assumptions," In what way do the other approaches? A bit more of discussion on this important point would be useful.

Approaches using equilibrium MaxEnt models (Refs. 34, 35, 41) start off with a physical polymer model with, for example, a bending stiffness and excluded volume interactions. This physical polymer model is then extended with additional effective interaction energies between genomic regions that are constrained by Hi-C scores using a MaxEnt procedure. These models are thus a hybrid between a physical polymer model and a maximum entropy model. The result of this procedure is a Boltzmann distribution for the conformations of the chromosome set by a Hamiltonian describing the full system, which comprises bending terms, soft- and hardcore repulsions and constraint-derived effective interaction energies between genomic regions. One of the advantages of such a description is that it makes concrete predictions for the polymer statistics at and below length scales where Hi-C derived constraints become relevant. However, such a description is only valid under the assumption that the chromosome can be described as an equilibrium system.

We made the following addition on page 2 (lines 154-162) to clarify this point:

"Importantly, although a mapping can be made to a statistical mechanics model, our approach does not rely on the chromosome being in thermal equilibrium. This is in contrast to approaches used in [34,35,41] where a hybrid MaxEnt procedure is employed combining a physical

polymer model with Hi-C derived constraints, resulting in an energy landscape description of equilibrium chromosome configurations."

1.5) Page 2: Unclear "a polymer on a 3D cubic lattice, with a subset of monomers representing N genomic regions. " What is "genomic region" here? What is the dimension of the lattice relative to the mentioned 10^5 constraints? What is the dimensionality of the σ ? In general, I got stuck for a while trying to make it absolutely clear to myself how the authors really represent their circular polymer (since there are many ways to represent its configurations), and being very slow and pedestrian here in the setup would help. Maybe a small schematic with a 3D lattice and a polymer inside; also mention that by construction of your ensemble and its moves you will ensure the polymer is always connected and circular (else one could incorporate these features as hard maxent constraints with infinite coupling in H, which I initially looked for but didn't find). In short, explain more clearly how the polymer is represented.

Thanks for bringing this to our attention. The polymer is represented as a discrete circular chain of length N on a 3D cubic lattice; the chain can self-intersect, but all configurations must be contained within the cell-shaped confinement. A subset of all N monomers, equally spaced on this chain, represent the centers of the genomic regions, which are defined as the stretch of the DNA associated to an individual bin of the Hi-C map. This defines the phase space, σ , i.e. the set of microstates in our system. In addition, for our polymer representation we need to specify the total number of monomers of the polymer and the lattice spacing, which we set using experimental data

More specifically, in our polymer representation, each 4th monomer represents the location of the center of a genomic region. The genomic regions are defined by the coarse-graining of the Hi-C map, consisting of 405 bins of ~ 10 kb - each of these bins we refer to as a genomic region. Thus, the total number of monomers we use is $N=1620$. The reason for choosing each 4th monomer on the chain to represent the center of a genomic region, rather than e.g. every single monomer, is that we can adequately capture our experiments on the measured distribution of spatial distances between subsequent genomic regions (SI Fig. S2), without being sensitive to the discrete lattice nature of our polymer representation. This procedure results in a lattice spacing of 88 nm. The cellular confinement then corresponds to a cylinder capped by spherical domes, containing 470 lattice points (more details on this are provided in SI section S2.2). To further illustrate our model setup, we followed the reviewers' suggestion and included an additional supplementary Fig. S4.

To obtain a rough estimate the dimensionality of phase space σ , we here provide a simple argument. In the absence of confinement, the number of possible states is equal to 6^N (in our case 6^{1620} with an associated entropy of 2903 nats). We can make a rough estimate of the confinement entropy using a simple blob argument. The largest contribution comes from the short-axis confinement in the cylindrical cell (radius $a=4.5 l_0$). This confinement introduces a blob size a , below which the scaling remains ideal, such that $a^2=gl_0^2$, with g the numbers of monomers in a blob. There will be Nl_0^2/a^2 of such blobs, each contributing $1 k_B T$ confinement free energy (or 1 nat in confinement entropy). Thus, we estimate a confinement entropy of 82 nats. To take the circularity of the polymer into account, we assume the locations of the endpoints to be uncorrelated within the confinement, which is reasonable as long as the confinement dimensions are much shorter than the polymer length. Approximating the endpoints to occupy each lattice site equally often, circularity reduces the number of microstates by a factor equal to the number of lattice sites in the cellular confinement (470 in our case, with an associated entropy of 6 nats). Thus, we estimate the number of microstates in σ to be roughly 10^{1222} . This dwarfs the dimensionality of the 10^5 constraints.

We made the following change to the main text to clarify our polymer representation (page 2 lines 107-116)

“the polymer is represented as a discrete circular chain on a 3D cubic lattice; the chain can self-intersect, and all configurations must be contained within the cell-shaped confinement. A subset of all N monomers---equally spaced along this chain---represent the centers of the genomic regions, which are defined as the stretch of the DNA associated to an individual bin of the Hi-C map. Thus, the dimensions of the coarse-grained representation are set by the resolution of the available Hi-C data (SI S2, S3.1).”

Finally, you are indeed correct that we use Monte Carlo moves that maintain the connectivity, circularity, and cellular confinement of the chain. In revised SI section S3.1 (with a new additional supplementary Fig. S4) we now mention explicitly that our Monte Carlo move set maintains the connected and circular nature of the chain, and ensures that the polymer remains within the cellular confinement.

1.6) Broken reference after equation S4 in the SI.

Thanks for catching this, it is fixed now.

1.7) Where does Eq 6 comes from -- what is the motivation for removing degeneracy due to unknown c by Eq 6? SI S3.2 does not explain, since it starts by assuming Eq 6 to derive what the energy shift should be.

Indeed, the derivation of Eq. 6 was not very explicit: a section has been added to S3.2 providing all necessary details.

1.8) Do you need to imply any regularization for epsilons? How come that pairs for which $f_{ij} = 0$ are not assigned epsilon \rightarrow infinity? I could only see this happening either because (i) the algorithm does not fully converge in epsilon space (although it may have converged until tolerance in the constraint space) OR, more interestingly, if (ii) the polymer nature of the problem (= loop) regularizes epsilons "automatically"...

We do not imply any regularization for the epsilons. Pairs for which $f_{ij} = 0$ are indeed effectively assigned epsilon \rightarrow infinity, which is mentioned in the caption of Fig. 1. We have now used a modified filter on the Hi-C data (See comment 3 by reviewer 2). Consequently, pairs for which $f_{ij} = 0$ are less common than in the filtered input data we used previously, but they are still present. To further clarify the treatment of pairs of genomic regions for which $f_{ij} = 0$, we also included the following in the SI after Eq. S1:

“For pairs of genomic regions i, j for which $\tilde{f}_{ij}^{\text{expt}} = 0$, the corresponding ϵ_{ij} is set to a high value at the start of the simulation, typically 10, which may further increase during iterations of the inverse algorithm. This initial value is high enough in practice to ensure these contacts do not form.”

1.9) You may consider citing De Martino et al, "Statistical mechanics for metabolic networks..." Nat Comms 9: 2988 (2018), along with other examples of maxent applications to biology. As in your case, there too sampling needs to explore a constrained space (as in your case, for the circular connected polymer), so special moves in the flux space are devised to be consistent with constraints.

Thanks for bringing this work to our attention, which indeed is very relevant. This reference is now cited after

“To this end, we build on existing MaxEnt methods for analyzing biophysical data”

Lines (82-84) on page 2.

Additionally, the following sentence was changed in SI section 2.2:

"The phase space of chromosome states is restricted to those that fit inside a cell, the sampling thus explores a constrained space (see also[9])."

Gasper Tkacik

Thanks, we really appreciate these constructive comments and suggestions, which helped us improve our manuscript.

Reviewer #2 (Remarks to the Author):

In “Learning the distribution of single-cell chromosome conformations in bacteria reveals emergent order across genomic scales”, Messelink and co-workers develop a maximum entropy approach to reveal aspects of spatial chromosome organization from Hi-C data as input. First, microscopy experiments are used to calibrate coarse-grained polymer simulations at short (<10 kb) length-scales. Then, using their iterative model optimization approach (MaxEnt), the authors infer interaction coefficients (ϵ_{ij}) between chromosomal loci which defines the maximum entropy configurational ensemble of chromosomal states. This ensemble of chromosome states is used to make testable predictions about the spatial positioning of loci within the bacterial nucleoid of *Caulobacter crescentus*. The approach is validated by comparing their inferred 3D chromosome positions to previously published FROS and FISH imaging data, showing good agreement. Among the interesting new findings, the authors find that local chromosome extensions and exclusions are present in the 3D spatial positioning inferred from their model; these are nice hypotheses that can be tested experimentally in the future.

The authors make an important contribution to the field of chromosome biology. This work will both be of interest to data scientists and theorists studying chromosome structure, and to a wider audience of biologists, as the authors generate testable hypotheses for the spatial positioning of specific chromosomal loci. Importantly, the method developed here, while demonstrated for *Caulobacter crescentus*, is readily generalizable to eukaryotic genomes as well, and may open new avenues for generating testable hypotheses in other species of bacteria with potentially different nucleoid geometries and chromosome organizations.

Overall, the method presented herein represents one of the most sensible (non-hypothesis driven) treatments of the “inverse problem” of chromosome structure elucidation from Hi-C data. The approach relies minimally on assumed distributions of spatial distances (except for the setting the coarse-graining length scale, which can be otherwise inferred), or specific detailed knowledge of the positioning of chromosomal loci as inputs. Moreover, this approach does not provide a ‘single representative chromosome structure’ as in some methods in the literature; they further introduce the notion of localization information which helps to formalize our understanding of chromosome positioning uncertainty and may help towards dispelling a prevalent conception of the ‘chromosome structure’ as static, or highly ordered structures. All combined, these features make the approach proposed here by Messelink et al. potentially quite valuable to the scientific community at large.

I recommend this article for continued consideration at Nature Communications. However, I recommend several important revisions to augment the rigor and clarity of the study before publication. Outlined below are my concerns, as well as some minor comments.

We thank the reviewer for commenting on the value of our work and for making a lot of helpful suggestions to improve the clarity and substantiate the rigor of our approach.

Major comments

2.1) The article main text is well written and generally clear. However, it is largely missing a discussion of the limitations of the MaxEnt method. It seems that some of the most important assumptions of the method are largely unstated, and readers are left to figure out what these are. Some suggestions of things to discuss include the assumption of no excluded volume on the polymer chain [see point 6]. Please also elaborate on how multiple-fork replication,

multiplicity of chromosomes, and poorly mapping regions in Hi-C will affect your results. For transparency and as a service to the readers, I strongly recommend that the authors highlight these (and other) limitations to their method in the main text and discuss their implications.

We thank the reviewer for raising this point, and we agree that the limitations and underlying assumptions of the MaxEnt method should also be clarified in the main text. The following assumptions are made in the MaxEnt approach:

- The MaxEnt model assumes that the Hi-C data (with only two-point contacts as opposed to three or higher order contacts) contains enough information to constrain the distribution of chromosome configurations, and the predictive power of the model will depend on this assumption. We present supporting evidence for the predictive power of our MaxEnt model through an independent test with localization data from microscopy experiments (Fig. 2).
- In our implementation of the MaxEnt model, we assume that contacts between genomic regions only form if the centers of the regions are within the same unit cell, which has a diameter of 176 nm (twice the lattice spacing). In addition, we consider the full ensemble of all coarse-grained circular polymer configurations, including self-intersecting configurations, in our MaxEnt model. Note, this does not imply that the microscopic configurations of the DNA can self-intersect. The inference approach selects for effective two-body interactions, which may in part represent excluded volume interactions at this coarse-grained level, to find the least-structured distribution of all polymer configurations.
- We measure the distribution of distances between loci 10kb apart to set the coarse-graining of the polymer. This measurement is done for 5 pairs of genomic regions, which all show similar distance statistics. We assume that *all* pairs of subsequent genomic regions follow these same distance statistics, although it could be that there are deviations from this along the chromosome.
- Our approach assumes a fixed cell size.
- We model a single, unreplicated chromosome, and assume that the Hi-C data is completely generated from such single, unreplicated chromosomes. Our method could be extended to replicating chromosomes, including multi-fork replication, which is ongoing work in our group. However, such an extension is outside the scope of this manuscript.

Other limitations of our approach are:

- The MaxEnt model does not make any statements about dynamics of the system: it is limited to describing the statistics of system states.
- The MaxEnt model is limited by the resolution of Hi-C data. The data set presented here divides the chromosome into bins of 10kb, thus any organizational features below this scale cannot be probed with our method. However, this can become possible as the resolution and accuracy of the data increases.

To further clarify these assumption and limitations, we made several changes to the manuscript.

On page 2 (lines 101-104):

“A central assumption of our approach is that the experimental Hi-C maps contain sufficient information to constrain the distribution of chromosome conformations.”

On page 2 (lines 107-116):

“the polymer is represented as a discrete circular chain on a 3D cubic lattice; the chain can self-intersect, and all configurations must be contained within the cell-shaped confinement. A subset of all N monomers---equally spaced along this chain---represent the centers of the genomic regions, which are defined as the stretch of the DNA associated to an individual bin of the Hi-C map. Thus, the dimensions of the coarse-grained representation are set by the resolution of the available Hi-C data (SI S2, S3.1).”

On page 8 (lines 531-542):

"Our approach resides in the class of static Maximum Entropy approaches, which make no assumptions or predictions about the underlying dynamics, as opposed to dynamical maximum entropy models or maximum caliber models (see for instance [52,53]). Further model limitations are set by the available input data: organizational features that cannot be faithfully encoded in population-averaged Hi-C data might be absent in the MaxEnt model. The resolution of Hi-C data is limited to 10kb for the data sets analyzed here, implying that any organizational features below this genomic length scale cannot be explored with our model.."

and on page 8 (lines 546-548):

"Furthermore, our approach may be generalized to other prokaryotes, including systems with replicating chromosomes and multiple replicons"

In the Methods:

*“As a cellular confinement, we use a cylinder capped with hemispheres with the dimensions of a newborn swarmer cell minus the cell envelope: $0.63 \mu\text{m} * 2.2 \mu\text{m}$ (SI S1-2), which is assumed to be the same for all cells”*

and the following to SI S2.2:

"The polymer model is allowed to intersect, since multiple centers of genomic regions could reside in the same unit cell volume. In fact, two monomers are defined to be in contact with probability γ if they simultaneously occupy the same lattice site. This assumes that the dominant contributions to Hi-C scores between two chromosomal regions are from configurations where their respective centers occupy the same unit cell. Any excluded volume effects reducing the number of self-overlaps of the coarse-grained polymer manifest as the effective two-body interaction energies, which are inferred during the Maximum Entropy procedure to satisfy the Hi-C constraints."

2.2) Reproducibility, code and data availability: the proposed MaxEnt method is overall pretty complicated to a non-expert as it involves 3D polymer simulations for the forward algorithm, and an iterative updating scheme for the inverse algorithm. I strongly recommend the authors make (a) the source code available, with some examples on how to run it, and (b) make data available for their MaxEnt generated conformations. The former will be important to minimize the barrier to entry for others looking to employ these methods, and the latter will be important for the wider community (including this reviewer) to more carefully evaluate the results of the author's modelling on their own.

We agree with the reviewer that it would be useful for the community to have access to our code as well as an ensemble of MaxEnt generated configurations. We will make both of these available upon publication.

2.3) **Data treatment:** raw Hi-C data is “cleaned up” (as described in Supplemental Section S5) to remove what the authors deem to be an “artefact” in the data (i.e. increased frequency of ori-ter contacts); however, the cleaning up procedure does not appear statistically or physically valid, and also relies on a number of subjective choices. I do not believe this treatment of the Hi-C data is acceptable for the following reasons:

a) The so-called artefact that is corrected for/removed (i.e. a higher frequency of ori-ter contacts than other similarly separated loci) is not really an artefact. These data points likely represent a real biological phenomenon arising from the active motion of the origin towards the terminus, and thus are an integral part of the biological ensemble of chromosomal states for that population of cells. The removal of interaction pairs below a certain frequency hinges on an interpretation (not solid data) of what is going on (i.e. it is indirectly inferred by Hi-C and microscopy) and does not seem scientifically sound.

Our aim in this paper is to investigate the Hi-C data of synchronized newly born swarmer cells. Ideally, these experiments would only include cells with a single (non-replicating) chromosome. In practice, however, there likely is a fraction of cells in which processes such as chromosome replication and segregation have initiated, which will be reflected in the Hi-C map. We agree with the reviewer that the *ori-ter* contacts observed in the Hi-C map are indeed physical, and they are likely due to the biological phenomenon of the replicated origin actively moving towards the terminus, as discussed in (Umbarger et al., 2012, Yildirim et al., 2018).

The data-processing procedure we employ assumes that these *ori-ter* contacts are exclusively due to newly replicated origins moving to the *ter* side of the cell. The low contact frequency between *ori* and *ter* is consistent with a small fraction of cells having initiated replication despite synchronization to the swarmer phase. The localization of the origin and terminus to opposite cell poles for an unreplicated chromosome, and the movement of the newly replicated *ori* to the terminus side are described in (Viollier et al., 2004). The central assumption underlying our data-processing analysis was also invoked previously in (Umbarger et al., 2012, Yildirim et al., 2018) to motivate neglecting the contact frequencies between *ori* and *ter* regions in finding chromosome conformations.

However, we agree with the reviewer that the treatment of these *ori-ter* contacts is better founded if it is largely data-driven. Therefore, we have now made use of Hi-C data sets for *C. crescentus* swarmer cells where replication is inhibited prior to synchronization, published previously in (Le et al. 2016). Indeed, in these experiments such an increase in contacts between the *ori* and *ter* genomic regions is not observed, further supporting our interpretation of these contacts. In our revised manuscript, we adapted our filtering procedure, making it more rigorous, and using the data sets for replication-inhibited cells as a benchmark. The new filtering procedure is detailed in new SI section SI S5.

The advantages of filtering the wild-type dataset, rather than applying the analysis only to the replication-inhibited cells, are two-fold: first, the experimental procedure to inhibit replication might affect features of chromosome organization. Second, a filter method allows for the analysis of data sets for mutants and cells in atypical growth conditions but without replication inhibition, such as the Δsmc mutant and the rifampicin-treated cells in (Le et al., 2013) using a single chromosome model. For completeness, the results of applying the MaxEnt method

directly to the unfiltered wild-type data, as well as to the replication-inhibited cell data, are presented in SI sections S7 and S8. Importantly, we find that all the central conclusions drawn in the main text based on our MaxEnt model trained on the filtered WT data, can also be drawn for a MaxEnt model on the unprocessed Hi-C data from the replication- inhibited cells.

To emphasize that our model is for a single, unreplicated chromosome, we added the following sentence to page 3, lines 201-203:

"Such newborn swarmer cells only have a single chromosome, whose replication has not yet initiated [43]."

We also added the following to the Methods section in the Main text:

Here, we consider Hi-C data (replicate 1 of the BglIII Hi-C data) on C. crescentus newborn swarmer cells published in [27], which have a single, non-replicating chromosome. However, due to imperfection synchronization, a small fraction of cells are included in these experiments in which processes such as chromosome replication and segregation have initiated, which will be reflected in the Hi-C map [28,47]. Before inferring a MaxEnt model, we apply a data-processing scheme to filter out contributions from cells with replicating chromosomes (See SI S5, S6). However, we also provide a MaxEnt model inferred directly from the unprocessed Hi-C data (See SI S7) and for MaxEnt models inferred from Hi-C data sets for replication-arrested cells [26] (See SI S8). While there are small differences between the different models, the central behaviors from the MaxEnt model reported in the main text is similar in all cases.

Finally, we significantly revised SI section 5 (Hi-C data filtering) and added new SI sections 6 (wildtype replicates), 7 (MaxEnt model on unfiltered Hi-C data), and 8 (MaxEnt model for Hi-C data on replication-inhibited cells) to address this point.

b) On a more subjective note, this filtering is motivated using “outside knowledge” about the system, which is contrary to the spirit in which the authors frame their method (i.e. the claim that the inference is entirely Hi-C data-driven).

The goal of the MaxEnt method is to derive a model for chromosome organization directly from Hi-C data, without making further assumptions about organizational features. In this paper, we present a model for the spatial organization of a single bacterial chromosome. This requires that the Hi-C data used as an input for the model is also taken for a single chromosome. Sufficient outside knowledge is thus needed to ensure the appropriate input data is used.

While the synchronization step used in (Le et al. 2013) results in the majority of swarmer cells containing a single chromosome, a small fraction of cells is expected to have initiated replication due to (see point a). This gives rise to small yet significant contributions in the Hi-C scores between *ori*- and *ter*-proximal regions. To filter out this effect, we made use of Hi-C data on replication-inhibited cells as a benchmark (see point a), minimizing the required outside knowledge to use wild-type Hi-C data for the MaxEnt model. However, we also apply our MaxEnt approach to unfiltered Hi-C data from replication-inhibited cells to further support our analysis and conclusions.

c) Most importantly, I wonder if the proposed data curation may have arisen from a misunderstanding of contact probability decay curves: in Region II where the data envelope is “fit” and points are removed does not appear to be noise. Instead, the “upwards trend” of the contacts beyond 2 Mb naturally arises when the authors perform log-binning of genomic

coordinates for a circular chromosome. This same “upwards trend” is present even for the most naïve models of chromosome conformations (e.g. a purely Gaussian circular chain with no confinement) and it does not constitute “spurious noise” as the authors claim. I believe it is thus incorrect for the authors to fit these interactions to a power-law, calculate two standard deviations, and subtract the resulting interaction frequencies from the data. To remedy these issues, I strongly recommend that the authors repeat the analyses without removing these contacts, or otherwise demonstrate more rigorously how their proposed treatment is statistically and physically sensible.

An upwards trend for average contact frequencies for the largest genomic distances would indeed be seen for a Gaussian circular chain without confinement, if the genomic coordinate runs further than half the chromosome length. However, in our analysis only intra-arm Hi-C scores are considered, which means that the largest genomic distances included are half of the total genomic length. For a Gaussian circular chain, the contact frequencies will decrease monotonously with genomic distance within this interval. The observed scaling behavior could therefore not be explained by the circularity of the chromosome.

To investigate the effect of our data processing procedure on the model results and our main conclusions, we now reran our MaxEnt inference directly on the unprocessed Hi-C data. In fact, we learned a MaxEnt model for three Hi-C data sets from (Le et al., 2013): two replicates using the BGIII restriction enzyme and one from an experiment using NcoI. Note, the frequencies of *ori-ter* contacts vary between these data sets, being the highest for the data set used in our analysis so far, and around a factor 3 lower on average in the NcoI data set. We find that the central main-text results are unchanged (SI SX), with the most apparent deviation between model and experiment occurring for *ter*-proximal regions, which show a longer tail of localizations extending into the *ori* half of the cell in the unfiltered model. Our MaxEnt model assumes a single non-replicated chromosome, which will thus interpret these elevated Hi-C scores as *ori-ter* interactions within the same chromosome. The changed localization pattern of the *ter* region induces long-axis correlations throughout the chromosome (SI figure S10). However, if long-axis correlations are conditioned on the *ori* and *ter* regions occupying opposite cell halves, the correlation pattern presented in the main text is restored again (SI figure S10).

We have revised our filtering procedure and the associated description in SI section S5 to provide a more rigorous treatment of our data processing approach.

2.4) In the section, “MaxEnt Model of the *C. crescentus* Chromosome Quantitatively Captures Measured Cellular Localization”, the authors state that, “[They] orient cells by setting the *ori* pole in the cell-half containing *ori*”. In other words, from the inferred chromosome conformations derived by the MaxEnt method, the *ori*’s are consistently placed in the same cell half. Unfortunately, the authors seemingly do not apply this procedure to the polymer simulations that they use as benchmarks (i.e. specifically, the “random polymer” simulations), which gives rise to misleading results. To illustrate this point, in Fig 2 (top panel): the “random polymer” simulation average long-axis position is a constant 0.5 over the entire chromosome length. I would firstly not expect a flat distribution of points for the average long-axis positioning if the “orienting procedure” were systematically applied (e.g. for the *ori* positioning, I would expect to see it closer to ~0.25 (and not at 0.5)); this suggests that the authors are treating the MaxEnt conformations and other polymer simulation conformations differently, and gives a false impression for the relative goodness of their approach. I suggest that the authors revisit these comparisons and fix the analyses appropriately.

We thank the reviewer for pointing this out. We have now modified Fig. 2 to include the localizations of the four loci for an oriented random polymer, which indeed allows for a better comparison with the MaxEnt model.

2.5) Relatedly to point 4, the authors use two types of polymer simulations as benchmarks by which to compare the MaxEnt method. The comparisons are to the “tethered random polymer” & “random polymer” throughout the text. However, a more appropriate model is the “tethered random polymer with juxtaposed chromosome arms”, or a simpler “polymer with juxtaposed arms but no tethering” model. This comparison also seems to be a natural extension of their previous work, where they have developed such simulation models (e.g. as in Ref. 27, for the case of active SMCs and tethered origins); moreover, the models I suggest above are more representative of the collective body of knowledge, as it incorporates information that has been in the field for >5 years (i.e. the fact that chromosome arms are juxtaposed, and tethered to the ori). I recommend these additional comparisons to be made, at the very least in the supplementary materials.

Thanks for this suggestion. We agree that an ensemble of chromosome states with juxtaposed arms is an interesting additional comparison case. To create such an ensemble, we decided to learn a MaxEnt model directly using only the average long-axis positions as constraints (similar to Fig. 2), which is a simple way of imposing the juxtaposition of the two arms. This construction has the advantage that it yields a MaxEnt distribution of chromosome states given juxtaposed arms, which enables a fair comparison with the MaxEnt model constrained by Hi-C data. A comparison with a more microscopic model in which the chromosome is organized with SMCs is also an interesting direction to explore, but we decided not to include this here. Such a model requires several parameters that we do not yet know from experiments, which makes the comparison to the MaxEnt model difficult.

We included a comparison of the main text results in Figs. 2-5 to a polymer with juxtaposed arms in a new SI section S16. While the juxtaposed-arm model by construction shows a similar long-axis localization (Fig. 2) and associated localization information (Fig. 5B) as the full Hi-C MaxEnt model, this juxtaposed-arm model does not exhibit the intricate structure in long-axis correlations (Fig. 3) and local extensions (Fig. 5A) that we observed for the Hi-C MaxEnt model.

We added the following to the main text page 4 (lines 280-284):

"Importantly, such organization is absent for a model with a tethered origin not constrained by Hi-C data, (Fig.3 B lower right), as well for a model with juxtaposed chromosomal arms only constrained by linearly organized average long-axis positions (SI S16)"

2.6) In the analysis of Super Domains the authors use the language “excluded” where it perhaps could lead to misunderstanding: e.g. “We find that these intra-arm anticorrelations result from the spatial exclusion of large genomic clusters between the two chromosome arms.” Based on the unstated model assumption of no excluded volume interactions (from their lattice polymer simulations), no two loci can be truly “excluded” from one another. Exclusion may only occur in the statistical sense: I wonder if the authors can demonstrate that such exclusions arise in single cell conformations, or if these are just statistical averages present from the population of cells. The notion of “spatial exclusions” is also seemingly at odds with the alignment of the left and right replisomes as seen by Hi-C. I suggest the authors address whether these exclusions are present within individual chromosome conformations, and address how said exclusions are also consistent with the 3C contacts, and consider revising their language in the main text.

Our chosen formulation was inaccurate in this case. The interaction between SuDs of the two chromosomal arms is indeed not a strict volume exclusion, but rather a statistical tendency not to overlap spatially. To emphasize this point, we made the following changes to the main text:

Page 4, lines 287-289:

"We find that these intra-arm anticorrelations are associated with large high-density clusters of subsequent genomic regions, which we term Super Domains (SuDs)."

Page 4, lines 307-310:

"This exclusion behavior is expected to generate negative intra-arm anticorrelations for pairs of genomic regions with similar average axial positions."

Page 5, lines 361-363:

"Correspondingly, the anticorrelations between long-axis positions of chromosomal arms are much stronger for this mutant (Fig. 3C lower right)."

To further illustrate the statistical tendency of SuDs not to overlap spatially, we now provide computed density plots of chromosomes based on our MaxEnt model. These simulated density plots represent predicted microscopy images of single chromosomes in a living cell (new Figure Fig. 4d). In the computed density plots, the SuDs show up as clearly defined high-density regions, which we compare to new super-resolution experiments of single cell chromosomes that indeed exhibit similar high-density clusters (new Figure Fig. 4E). The computed density plots illustrate that SuDs from different arms tend to spatially exclude each other, which is also borne out of our more detailed correlation analysis discussed in section S9.1.

2.7) In Figure 2B, the single cell distributions of chromosomal loci by FROS do not seemingly match with the values plotted in Figure 2A. For instance, a locus tagged at the *ori* seems to have a scaled mean long-axis position of ~ 0.1 in Figure 2B but averages out to a position of ~ 0.2 in Figure 2A. Can the authors comment on the difference?

Along these lines, can the authors comment on how their MaxEnt model (and well as the tethered random polymer model) finds the "exact cell positioning" of the *ori* to be at 0.2 along the long axis? It seems rather non-trivial that this should be the default placement... is it externally imposed?

This is well spotted by the reviewer. Importantly, the distributions in Fig. 2B are a separate data set to the averages in Fig. 2A. MT, also co-author on the 2004 paper where these datasets were taken from, suggests the discrepancy between the data sets could be due to the image analysis methods used in this paper. The experimental averages presented in Fig. 2B were obtained with one of the first automated image analysis packages available. There may be some variations due to occasional detection issues resulting from focus problems, insufficient contrast of the phase contrast images, and related issues. Unfortunately, the authors of the 2004 Viollier paper could not recover all the raw data from their databases for re-analysis. Therefore, we simply took the data directly as published in the 2004 paper.

On the positioning of *ori*: for the MaxEnt model, this average position arises automatically after enforcing the Hi-C constraints, and it is not externally imposed. We agree with the reviewer that this is a non-trivial result. Thus, this predicted long-axis localization of the genomic regions (Fig. 2A) is a strong validation of the predictive power of a MaxEnt model based solely on Hi-C data. For the tethered random polymer, used as comparison case for our MaxEnt model, the

tether is chosen such that the average position of the *ori* is at ~ 0.2 along the long cell axis, thus in this case the *ori* position is externally imposed.

2.8) Supplemental section S2.2 is confusing and seemingly inconsistent with S3.1—more details are required. Specifically, it states “In this representation [of the polymer], the position of each even monomer indicates the cubic unit cell occupied by the center of a Hi-C chromosomal region”. This is seemingly in contrast with section S3.1, which states that “Each 4th monomer represents the location of the center of a genomic region, with three monomers in between to ensure Gaussian statistics between subsequent centers of genomic regions”. I was also wondering if the authors could elaborate on how their coarse graining leads to 7 possible distances (length $0b$, $\sqrt{2}b$... , $4b$); perhaps some illustrations/sketches may help the reader visualize where these are coming from.

Thanks for pointing this out. In response to the referee 1 (comment 1.5) we clarified the description of our model representation in both the main text and SI.

In our representation, each 4th monomer represents the location of the center of a genomic region. To address the concerns of the referee related to this, we modified the start of section S2.2 to the following:

"We employ a polymer model on a cubic lattice. In this representation, the position of each fourth monomer indicates the cubic unit cell occupied by the center of a Hi-C chromosomal region. The polymer model is allowed to intersect, since multiple centers of genomic regions could reside in the same unit cell volume. In fact, two monomers are assigned a contact probability γ only if they simultaneously occupy the same lattice site. This assumes that the dominant contributions to contacts between two chromosomal regions are from configurations where their respective centers occupy the same unit cell. Any excluded volume effects reducing the number of self-overlaps of the coarse-grained polymer manifest through imposed Hi-C score constraints.

To set the scale of the lattice spacing b in the model, we use the average spatial distance between consecutive Hi-C chromosomal regions determined in Sec. S2.1). If we consider distances between subsequent chromosomal regions, however, coarse-graining effects need to be taken into account: only seven distances between these regions are possible in the lattice representation (Fig. S3 B): $(0, 2b, 2b, 6b, 8b, 10b, 4b)$, which occur with respective relative occurrence frequencies (f_1, \dots, f_7) ."

Additionally, we included a new supplementary figure S4 to illustrate our lattice representation of a coarse-grained chromosome.

2.9) The authors develop a concept of localization information, which uses the ensemble of 3D polymer conformations to define the likelihood of being able to find a particular chromosomal segment in a pre-defined nucleoid compartment. It took me a while to appreciate the significance of this result. I think this is an important concept for the field moving forward, but it is also very surprising that the localization information is so low, with the highest localization information value of 3 (or the equivalent of localizing to 1/8th of the cell). This highlights the inherent stochasticity of 3D chromosome structures. I think the paper could benefit from a slightly more elaborate discussion of this subtle concept; perhaps the authors could give more examples to build reader intuition for the localization information (and its meaning) and to contextualize its biological significance.

Thanks for stressing the importance of this concept. We followed this suggestion and made several changes to this section to better explain the idea behind the localization information and also the value of 3 bits into perspective.

We made the following modifications on page 6, lines 416-429:

"This spatial information depends on the degree of localization of genomic regions. Put simply, the localization information content of a genomic region increases with the precision of its cellular location, i.e. when the spatial distribution of the genomic region is more sharply peaked around a specific point in the cell. This localization information could for example be used to localize proteins more precisely within the cell: a high relative affinity to a genomic region with a high localization information increases the localization of this protein in the cell. This mechanism may be exploited to position protein droplets [49], through nucleation on specific chromosomal regions, e.g. droplet-like clusters of DNA-binding chromosome partitioning proteins of the ParB family [3]"

And on page 6, lines 432-445:

"This chromosomal localization information is largest near ori and ter providing 3 bits of localization information, equivalent to reducing the localization uncertainty to one cellular octant. By contrast, a random polymer provides only 1 bit, enough to reduce localization uncertainty to one cell half. For comparison, with our coarse-grained description a maximal localization information of approximately 9 bits could be achieved. Thus, while this localization information metric indicates that the bacterial chromosome is substantially more ordered than a random polymer, it also highlights that the chromosome is far from a rigid organization with a precise folded structure. "

Minor comments

2.10) The authors claim their method is compatible with non-equilibrium dynamics, but do not elaborate on how. Please clarify this statement, since: 1) it appears that by defining the ϵ_{ij} as a temporally invariant quantity, or 2) in the absence of any limit cycles, or 3) active forces in the model, the proposed method is not really addressing the question of non-equilibrium dynamics...

Thanks for raising this point. The MaxEnt approach is designed to find the least-structured distribution of chromosome conformations consistent with experimental data, but the model does not make any assumptions or predictions about the underlying dynamics of the system. There are many dynamical models that could give rise to this (steady-state) distribution, including models that break detailed balance. The only assumption of our approach is that the Hi-C measurements are sufficient to constrain the distribution of chromosome conformations. While, the resulting distribution is parametrized as a Boltzmann distribution, this does not imply that the underlying physical dynamics of the chromosome has to obey thermal equilibrium dynamics.

This is in contrast to approaches using equilibrium MaxEnt models (Refs. 34,35,41), which start off with a physical polymer model with, for example, a bending stiffness and excluded volume interactions. This physical polymer model is then extended with additional effective interaction energies between genomic regions that are constrained by Hi-C scores using a MaxEnt procedure. These models are thus a hybrid between a physical polymer model and a maximum entropy model. One of the advantages of such a description is that it makes concrete

predictions for the polymer statistics up to length scales where Hi-C derived constraints become relevant. However, such a description is only valid under the assumption that the chromosome can be described as an equilibrium system.

We made the following addition on page 2 to clarify the distinction with equilibrium MaxEnt models (lines 154-162):

"Importantly, although a mapping can be made to a statistical mechanics model, our approach does not rely on the chromosome being in thermal equilibrium. This is in contrast to approaches used in [34,35,41] where a hybrid MaxEnt procedure is employed combining a physical polymer model with Hi-C derived constraints, resulting in an energy landscape description of equilibrium chromosome configurations."

Additionally, we included the following section in the discussion on page 7 to emphasize that our MaxEnt model remains agnostic about dynamics (lines 531-535):

" Our approach resides in the class of static Maximum Entropy approaches, which make no assumptions or predictions about the underlying dynamics, as opposed to dynamical maximum entropy models or maximum caliber models (see for instance [52,53])"

2.11) In page 2: "To this end, we introduce constraints to the entropy functional: [Eq. 2]. " Strictly speaking the equation is not for the constraints themselves but for the Lagrangian... a reference to the method would be helpful, and a tweak in wording. E.g. "To this end, we introduce constraints to the entropy functional by introducing the Lagrangian function: [Eq. 2]."

We agree with the reviewer that the wording can be improved here. We chose to now follow the wording also used in (Bialek, W. (2012). *Biophysics: searching for principles*. Princeton University Press.) and made the following change on page 2 (lines 135-137):

"To this end, we introduce the functional \tilde{S} , with one Lagrange multiplier λ_{ij} for each experimental constraint and λ_0 ensuring normalization:"

Additionally, we included a reference for the general MaxEnt procedure on page 2 (lines 127-130) as follows:

"To obtain the least-structured distribution of microstates consistent with experiments, we seek $P(\{\mathbf{r}\})$ that maximizes S (Eq[1] under experimental constraints[38,40]. The two constraints we impose are:"

2.12) A reference is needed for the mapping of Eq. 4 to the confined lattice polymer, Eq. 5.

The mapping is made between Eq. 3 and Eq. 5, we added a sentence after Eq. 5 to make this more explicit:

"The mapping to Eq.3 is made by setting $\epsilon_{ij} = \gamma \lambda_{ij}$, where ϵ_{ij} are the effective interaction energies between overlapping loci in the Hamiltonian formulation."

We have not added a reference because this mapping is straightforward and, to our best knowledge, has not been done before in this form. We do note that similar kind of mappings have been performed in many of the MaxEnt papers that we cite at the beginning of this section.

2.13) The supplement has a bibliography but in-text references are missing to these cited papers.

Thanks for bringing this to our attention, which may have been a compilation error. We made sure to include all in-text references in our resubmitted manuscript.

2.14) In section S2.1 what is the rough value for the “drift between frames” before the correction?

The estimated drift in the x direction is 35 ± 4 nm, the average drift in the y direction is 52 ± 5 nm (error of the mean).

We now mention this in S2.1

2.15) S2.1 is written in a slightly complicated and confusing manner. The descriptions can be simplified (e.g.). Moreover, it stated that the loci distances are Gaussian (page 8) referring to Figure S1. The distances are not Gaussian as they only have positive values...

We agree with the reviewer that S2.1 was not very accessible. We significantly rewrote this section to describe our procedure more clearly. Indeed, the distances are not Gaussian, it should be the 'relative positions'. This is now changed in the supplement on page 10.

2.16) More details are needed for the Forward simulation (Fig. S3.1): what are the initial conditions for the simulations? Are configurations equilibrated?

Thanks for bringing this to our attention. To address this, we added the following elaborations to the SI:

- On page 13, after *“The algorithm is initiated with the polymer randomly arranged within the confinement”*:

“This starting state is obtained by first 'winding up' the polymer in a square that fits in the confinement. Subsequently, a simulation with no interaction energies is run for 10^6 Monte Carlo moves. The resulting configuration is used as the starting configuration.”

- At the end of S3.1:

“At the start of the forward simulation, we apply a burn in time of 20×10^6 MC moves before contact frequency statistics are calculated. During the inverse algorithm, this burn in time is only applied to the first forward simulation. For subsequent forward simulations, the final configuration of the previous forward simulation is used as a starting state.”

2.17) Some equations are missing, likely in the LaTeX file compilation (supplement page 12, after Eq. S4).

Thanks, we fixed these compilation problems in our revised manuscript.

2.18) Sometimes it was a bit confusing what is being referred to (i.e. whether S12 is equation S12 or section S12 – e.g. page 9 of supplement). Please consider prefacing all the equations in with Eq. SX, or sections with “Section SX”. Note: section S9 also has references to Figs. SS12, and SS13 (extra S) as does Section S12 (SS18, SS19).

Thanks for bringing this to our attention, these ambiguities and extra 'S's have been resolved.

Finally, we thank the reviewer for all these constructive comments and suggestions, which helped us improve our manuscript.

Reviewer #3 (Remarks to the Author):

The manuscript describes a Maximum Entropy approach to reconstruct single-cell 3D structures of bacterial chromosome based on population-based Hi-C maps. The modelling part is done with a lot of rigor, and likely calculates the maximum entropy ensemble correctly.

The application of the method to the bacterial system is of potential interest; however, it is complicated by the fact that there are no known ways in which organization of bacterial chromosomes influences important cellular processes (e.g. gene expression).

This makes it difficult to evaluate and interpret results of the manuscript in a broader context. The presented model does not reveal any new mechanisms of chromosomal folding, or any influences of chromosomal folding on cellular processes.

Analyses of the model are not done very carefully. The authors “brush off” the fact that they see good agreement on one chromosomal arm and not on another. They do not provide any justification of the SuD domains or any single-cell data that would support them.

In general, I would much rather see applications of this method to mammalian systems. It would be interesting to see this model applied to other Hi-C maps, such as TADs in mammals, micro-C data with grids of loops (Krietenstein et al, 2019), etc. I strongly encourage the authors to consider these systems in this or subsequent manuscripts.

I believe that the model presented by the authors correctly infers a maximum entropy ensemble subject to the constraint of a Hi-C map. However, the model is only applied to the bacterial data and does not make any conclusions relevant to biology. Moreover, I have doubts about applicability of maximum entropy reconstructions to Hi-C data in general. Therefore, I believe that the manuscript is of limited interest to the broad scientific audience, and is better suited for a more specialized journal.

We thank the reviewer for commenting on the rigor of our approach and on the interest our work may have for bacteria. We agree with the reviewer that more knowledge exists about how chromosome organization impacts biological processes in eukaryotic cells. Nonetheless, ample examples exist in prokaryotes, where the chromosomal organization is shown to affect processes such as transcription, chromosome segregation and cell division. We thank the reviewer in pointing out that we should clarify this in our manuscript and we have followed their suggestion to do so (detailed below).

We share the interest of the reviewer to apply this approach to mammalian cells. It is our ambition to do so in future work, but mammalian cells are outside the scope of this manuscript. We feel that the application of our approach to bacterial cells is of sufficient general interest, which is also supported by the comments of the other reviewers. We feel that the distribution of single-cell chromosome conformations that we infer is highly relevant for biology. We show that we can predict features such as the cellular localization of genes over the whole chromosome quite accurately from our model. In addition, we make many new predictions such as the new chromosome density plots (revised Fig. 4E) and two-point correlation functions in Fig. 3, which provide a systematic way of characterizing structure in the distribution of chromosome conformations. In all cases, we show how our results are affected by biologically relevant perturbations such as rifampicin treatment (inhibits gene expression) and the Δsmc cells.

Finally, we followed the insightful suggestion of the reviewer to provide further experimental support for the existence of SuDs. Specifically, we now included super-resolution microscopy data (revised Fig. 4D), which we compare to new chromosome density plots from our model, including SuD associated structures (revised Fig. 4E). With all these improvements and additions, we believe that our revised manuscript is of interest to the readership of *Nature Communications*.

To further clarify the significance of chromosomal organizing for biological processes in bacteria, we made the following changes to the manuscript:

We expanded the sentence on page 1 (lines 22-25)

“Various proteins regulate bacterial chromosome structure [1-5], imposing order on its spatial organization.”

To

“Various proteins regulate bacterial chromosome structure [1-5], imposing order on its spatial organization and thereby impacting cellular processes such as transcription [6]”

New Ref.:

[6] Dorman CJ (2014) Function of Nucleoid-Associated Proteins in Chromosome Structuring and Transcriptional Regulation. *J Mol Microbiol Biotechnol* 24: 316-331.

We expanded the sentence on page 1 (lines 38-41)

*“Indeed, fluorescence microscopy experiments revealed that chromosomal loci localize to well-defined cellular addresses in various species [7, 15–17], including *Caulobacter crescentus* [18].”*

with

“This organization helps steer chromosome segregation [19] and cell division [20]. In addition, the level of transcription of several genes depends on their distance to the pole [21].”

New Refs:

[19] Toro E, Hong S-H, McAdams HH, Shapiro, L (2008) *Caulobacter* requires a dedicated mechanism to initiate chromosome segregation. *Proc Natl Acad Sci USA* 105: 15435-15440.

[20] Thanbichler M, Shapiro L (2006) MipZ, a spatial regulator coordinating chromosome segregation with cell division in *Caulobacter*. *Cell* 126: 147-162.

[21] Lasker K, von Diezmann L, Zhou X, Ahrens DG, Mann TH, Moerner WE, Shapiro L (2020) Selective sequestration of signalling proteins in a membraneless organelle reinforces the spatial regulation of asymmetry in *Caulobacter crescentus*. *Nat Microbiol* 5: 418-429.

In the discussion we rephrased the sentence (lines 524-530):

*“This information reaches up to 3 bits around *ori* and *ter*, equivalent to a localization uncertainty in the cell of one cellular octant. We speculate that such localization information encoded by the organization of the chromosome could be exploited for sub-cellular positioning of proteins and protein droplets or for regulation of transcription of genes, as was observed in [21].”*

Major comments

3.1) The result that areas of large extension correspond to active genes on one chromosomal arm is perplexing, and warrants much deeper investigation. However, the manuscript just “brushes it off” by suggesting some kind of symmetry breaking: “Thus a feature is required to break this symmetry.”.

I strongly disagree with this approach. I think the authors have to get down to the root of the difference between the two arms. The agreement of the peaks of local extension with the active genes seems rather good on the first arm; they may agree for the first couple peaks on the second arm, but then the agreement disappears.

There are few possible explanations. One of them is that highly expressed genes on the second arm do not produce a strong signature in the Hi-C maps (i.e. do not correspond to the peaks of insulation). Perhaps, the authors should compare insulation in Hi-C maps with local extension, and then insulation with highly expressed genes, to see which of the two relationships are broken. If the former, then it suggests a potential problem with the algorithm. Another explanation is that the relationship between local extension and insulation in Hi-C maps is non-trivial (e.g. having lots of insulation peaks on an arm does not produce much local extension). This could be studied by feeding synthetic Hi-C maps into the model. Finally, there could be a bug in the code that leads to gradual de-synchronization of the two plots.

The presence of the correlation between the genomic position of high local extension and location of highly transcribed genes for the right chromosomal arm is indeed a striking result. We have performed additional analysis to get to the root of the difference between the two arms, as detailed further below. However, the origin of this phenomenon cannot easily be deduced from the MaxEnt model. The MaxEnt model yields the distribution of single cell chromosome configurations, but does not provide a mechanistic description of the processes giving rise to this distribution. Thus, this result is presented as a prediction of the model and we think it is important to point out that the underlying mechanism is still unclear.

To address the reviewers concern about our original wording (“Thus a feature is required to break this symmetry.”) and to emphasize that this feature is a prediction of our model that awaits further exploration, we modified the following text on page 6 lines 400-409:

"If the colocalization of local extension peaks by highly transcribed genes would only depend on the relative direction of transcription and replication, this should also occur for highly transcribed genes on backward strands on the left arm, which we do not observe. Thus, while our results indicate a connection between high local chromosome extension and the direction of replication and transcription of highly transcribed genes, the underlying molecular mechanism is still unclear."

To further investigate the connection with regions of high insulation as suggested by the reviewer, we use the boundaries of Chromosomal Interaction Domains (CIDs) as defined in (Le et al., 2013) as a marker of such high insulation regions. Analyzing the coincidence of local extension peaks and CID boundaries, we find that these most strongly colocalize on the right chromosomal arm (Fig. R2B). For the left chromosomal arm, on the other hand, the colocalization is lower than would be expected by chance. Similarly, highly expressed genes most strongly colocalize with CID boundaries on the right chromosomal arm, whereas there is no significant colocalization on the left arm (Fig. R2D). We thus see that the connection between local extension peaks, CID boundaries and HTGs (Highly Transcribed Genes) on the forward strand is present exclusively on the right arm in all three comparisons. However, the correlation is strongest between local extension peaks and HTGs on the forward strand of the

right arm (SI Figure S22E). Thus, the difference between the left and right arm is not only present in our MaxEnt model predictions, it is already present in the Hi-C map.

These results suggest that the asymmetry between the left and right chromosomal arm is present in all comparisons proposed by the reviewer. This implies that the relation between local extension peak locations and CID boundaries is dependent on genomic position, and that HTGs do not give rise to CID boundaries on the left chromosomal arm. Further investigating this asymmetry using independent methods would be an interesting direction for further investigation.

Figure R2 Analysis of the degree of overlap between peaks in local chromosome extension, CID boundaries and locations of highly transcribed genes (HTGs). **A)** Green solid line: fraction of local extension peaks that coincide with the location of a CID boundary, as a function of the cutoff factor α . The dashed line indicates the expected fraction of overlap for randomly chosen locations of peaks, the light green area indicates the 95% confidence interval around this expected fraction. The grey line indicates the number of peaks included for a given cutoff factor (indicated on the right axis). **B)** The same analysis as in A), performed separately for the right (0-2 Mb, blue) and left (2-4Mb, red) chromosomal arms. **C)** Green solid line: fraction of CID boundaries that coincide with the location of a HTG. The dashed line indicates the expected fraction of overlap for randomly chosen locations of CIDs, the light green area indicates the 95% confidence interval around this expected fraction. **D)** The same analysis as in C), performed separately for the right (0-2 Mb, blue) and left (2-4Mb, red) chromosomal arms.

3.2) The anti-correlation between the two chromosomal arms is an interesting result. I believe that it has a plausible biological explanation, however, a mechanistic model such as the one in (Le, 2013) would be needed to test mechanistic hypotheses. Normally, tethering ori at one end of the nucleolus aligns the two arms together. However, if the ori were to be attached not strictly

at the pole of the nucleolus, one of the arms would fill the ori end of the nucleolus, and then proceed towards the terminus side of the nucleolus. Another arm would go towards the terminus right away due to steric exclusion. The whole chromosome would “rotate” with (ori – epsilon) or (ori + epsilon) being where ori should be theoretically. This would lead to anti-correlation between positions of the two arms.

It would not arise in a long polymer considered as a control because it is too long. However, I believe that the effect would arise naturally if the authors were to compute similar correlations in a circular bottlebrush model as in (Le 2013). A mechanistic explanation of why this anticorrelation arises, and what are the biological implications of it, would be very helpful for understanding the results of the model.

We thank the reviewer for expressing interest in these results and for these insightful comments. While a global rotation would indeed induce anticorrelations between juxtaposed regions, this would also induce additional anticorrelations between other genomic regions on opposite chromosomal arms that are not observed in the MaxEnt model.

To illustrate the features of a long-axis correlation map that would be induced by a global rotation, we simulated the effects of such rotational fluctuations. Specifically, we took a set of configurations from our model, and generated an ensemble of new configurations by performing a rotational fluctuation with a random magnitude of all genomic regions along the polymers axial coordinate within each configuration. The magnitude of this rotation was drawn from a zero-average normal distribution, with the standard deviation σ treated as a free parameter. For this new ensemble of configurations, including global rotation fluctuations, the long-axis correlations were calculated between all genomic regions. The resulting long-axis correlation maps for this rotational model for four choices of the standard deviation are shown in Fig. R3.

We see that for $\sigma=0.2$ Mb, the magnitude of correlations in the rotation model (Fig. R3A, upper left) is comparable to those observed in the original MaxEnt model (Main text Fig. 3B, upper left). Importantly however, the anticorrelations in the rotation model are present between *all* genomic regions on opposite stretches of the chromosome. Thus, in this case, we see anti-correlation both between opposing genomic regions on the left and right chromosome arm and between opposing genomic regions near *ori* and *ter*. This is in contrast to the pattern observed in the original MaxEnt model, where the anticorrelations are only present between juxtaposed genomic regions lying on opposite sides of the left and right chromosome arms and opposing genomic regions near *ori* and *ter* exhibit positive correlations (Main text Fig. 3B, upper left). For larger values of σ , the anticorrelation pattern in the rotation model initially remains qualitatively the same as for low σ , but the magnitude of correlations increases (Fig. R3A, lower right). For even larger values of σ , the long-axis correlation pattern starts to qualitatively change: the region of anticorrelation between *ori* and *ter* becomes larger (Fig. R3B). Furthermore, the magnitude of anticorrelations is much higher for these values of σ than observed in the original MaxEnt model.

Figure R3 Long-axis correlations for chromosome configurations with global rotational fluctuations. A) Upper left: long-axis correlations for model configurations with global rotational fluctuations along the polymer axis, drawn from a normal distribution with $\sigma=0.2\text{Mb}$. Lower right: the same for $\sigma=0.3\text{Mb}$. B) Upper left: same for $\sigma=0.7\text{Mb}$, lower right: same for $\sigma=1\text{Mb}$.

We added a new supplementary section S12 discussing long-axis correlations arising from a global rotation.

We agree with the reviewer that it would be interesting to study the correlation pattern emerging from a circular bottlebrush model of the bacterial chromosome, such as the one presented in (Le et al., 2013). While we don't have the Le model to our disposal, we can construct a model with similar features using a modified MaxEnt approach. Specifically, we decided to learn a new MaxEnt model using only the average long-axis positions as constraints (similar to Fig. 2), which is a simple way of imposing the juxtaposition of the two arms. We included a comparison of the main text results in Figs. 2-5 to a polymer with juxtaposed arms in a new SI section S16. While the juxtaposed-arm model by construction shows a similar long-axis localization (Fig. 2) and associated localization information (Fig. 5B) as the full Hi-C MaxEnt model, this juxtaposed-arm model does not exhibit the intricate structure in long-axis correlations (Fig. 3) and local extensions (Fig. 5A) that we observed for the Hi-C MaxEnt model.

While a global rotation or the linear juxtaposed organization of the chromosome do not explain the observed long-axis correlation structure, we do offer an explanation that accounts for this pattern. We find that in individual chromosomal configurations, large high-density clusters of subsequent genomic regions emerge, which we term Super Domains (SuDs) (Main text page 5 and Figure 4). We find that SuDs that form on opposite chromosomal arms tend to spatially exclude each other. As a result of this tendency to spatially exclude, chromosomal regions belonging to SuDs on opposing sections of the two arms, are expected to fluctuate in an anti-correlated fashion. In the revised Figure 4, including the new measured and computed chromosomal density plots in panels D and E and the associated revisions in the main text, we further emphasized and clarified this point. In future work, we will further explore the mechanistic origins how SuDs affect long-axis correlations.

3.3) The authors claim to have discovered a new kind of domains in single cells, termed SuDs. Those domains appear only as a product of the model the authors present. The authors do not establish any connection between SuDs and other biological data, and do not verify SuDs by any single-cell experiment.

The bioinformatics analysis of SuDs is equally lacking. Are SuD boundaries enriched in something? Can you predict them from biological data? What are average positions of boundaries? Can you see them in microscopy?

Even if these statistics are presented, I would still take SuDs with a grain of salt, because the authors have no direct single-cell measurements of the domains that they discover. Other single-cell domains were met with a degree of controversy in the field despite being observed directly in single-cell experiments (“Contact domains” in Flyamer et al, 2017; and “TAD-like structures” in Bintu et al, 2018 come to mind).

We agree with the reviewer that direct microscopy tests of the presence of the predicted SuDs are important. We thus followed their excellent suggestion to investigate the presence of SuDs with single-cell experiments. We employed SIM (Structured Illumination Microscopy) super-resolution microscopy and investigated the intracellular distribution of the chromosome in living *C. crescentus* cells at the single cell level. These experiments reveal that the chromosome exhibits a highly heterogeneous spatial distribution in the cell, including several dense cluster-like regions (new Main Text Fig. 4D). We observe that the number, size and location of these high-density regions vary from cell to cell, consistent with SuD properties derived from our MaxEnt model. Similar blob-like structures have also previously been observed with (super-resolution) microscopy for the chromosome of *Bacillus subtilis* (Marbouty et al, 2015) and *Escherichia coli* (Wu et al., 2019), suggesting that SuDs are also present in other bacteria.

To compare these single-cell experimental results with theory, we provide computed density plots of chromosomes based on our MaxEnt model. Specifically, for each chromosome configuration in our model, we compute a chromosome density plot for a slice along the length of the cell. A Gaussian blur is applied to this density plot, such that the resolution in the z -direction (300 nm) and in the x,y directions (120 nm) are set to match the experimental resolution. In the computed density plots, the SuDs stand out as high-density regions. The comparison between microscopy results and MaxEnt configurations is presented in new main text figure panels Fig. 4D,E. In summary, in the computed density plots, the SuDs show up as clearly defined high-density regions, which we compare to super-resolution experiments of single cell chromosomes that exhibit similar high-density clusters. These new results allow us to establish a connection between the SuDs predicted by our model and single cell super resolution experiments. Thus, these new results lend further support to the existence of SuDs in the *C. crescentus* chromosome. Further work to investigate the SuD properties in live cells is planned in a future study.

We made the following changes to the main text to discuss these new results on page 4 lines 311-334:

"To experimentally verify signatures of SuDs, we turned towards SIM (structured illumination microscopy) super-resolution microscopy and investigated the intracellular distribution of chromosomal DNA in C. crescentus at the single-cell level. These experiments reveal that the chromosome exhibits a highly heterogeneous spatial distribution in the cell, including several dense cluster-like regions (Fig4D). We observe that the number, size and location of these high-density regions are found to vary from cell to cell, consistent with SuD properties derived from our MaxEnt model.

To compare these single-cell experimental results with theory, we provide computed density plots of chromosomes based on our MaxEnt model. Specifically, for each chromosome configuration in our model, we compute a chromosome density plot at the experimental

resolution (see Methods), as shown in (Fig. 4E). In the computed density plots, we observe high-density regions similar to those obtained in our super-resolution experiments. Importantly, the high-density regions in the modelled chromosome density plots correspond to underlying SuD structures (dashed lines in Fig 4E). Thus, these new results allow us to establish a connection between the SuDs predicted by our model and single-cell super-resolution data."

And on page 7, lines 496-500:

"Similar blob-like structures have previously been observed with (super-resolution) microscopy for the chromosome of *Bacillus subtilis* [24] and *Escherichia coli* [13] suggesting that SuDs are also present in other bacteria."

The details of these new SIM super-resolution experiments are presented in new SI section SI S 1.2.

We agree with the reviewer that it would be interesting to further explore SuD features with the MaxEnt model. We investigated several additional properties of SuD structure and organization, which are presented in supplementary figure SI Fig. S9.2 (See also Figure R1). Importantly, as the SuDs are not localized on average (main text Fig. 4C), the boundaries are not linked to specific genomic regions. We do however find that the average cluster size and the average positions of cluster centers vary as a function of genomic position (SI Fig. S21D,E). We find the average SuD size to be largest close to the *ori* and *ter* regions (SI Fig. S21D), and the probability of a SuD appearing to be largest in these regions as well (SI Fig. S21E). Diving deeper into SuD properties would be very interesting, and we are in the process of addressing this question in future work.

3.4) On a broader scale, I have doubts about the validity of maximum entropy methods to infer single-cell conformations. In actual cells, folding of chromosomes is a consequence of a fairly complicated and potentially active process involving many architectural proteins. How can we be sure that the conformational ensemble achieved by this process matches the maximum entropy ensemble?

Imagine that all you have is an average 2D projection of a building (or many buildings). With a maximum entropy approach, a reconstruction of the ensemble of buildings, subject to the constraint of this projection, would uniformly fill the space between the ground and the roof with uniformly-distributed building materials: windows/walls on the outside, walls/doors in the center. In reality, a building was obtained through a particular process of making it, that put building materials into walls and floor, and left rooms empty. This reduced the entropy, which is expected for any building process. Same logic could be applied to the enzymes that fold chromosomes: they don't have to follow the maximum entropy ensemble; they likely reduce entropy and achieve only a particular subset of this ensemble. The MaxEnt method derives a most general potentially non-equilibrium set of conformations matching the constraints, while the reality is likely a not-so-general non-equilibrium set of conformations matching the constraints.

If one were to apply MaxEnt approach to maps of mitotic or prophase chromosomes, would they recover the nested condensin I/II loops with condensins forming a scaffold in the center? My guess would be that they won't: the ensemble created by making the loops-on-the-scaffold structure is too constrained, and occupies a tiny sub-space in the whole maximum entropy ensemble consistent with fairly featureless mitotic Hi-C maps.

One of the central assumptions of our MaxEnt model is that the Hi-C data contains enough information to constrain the distribution of chromosome configurations (which we now clearly state in the main text – lines 101-104). Note, the Hi-C map contains information about how proteins shape the chromosome, which is illustrated by the difference between the wildtype, Δsmc , and rifampicin treatment data sets. Our approach does not find a maximum entropy model for a circular polymer in the confinements, which would be a random confined polymer, but a maximum entropy model for a circular polymer in confinements that has to reproduce the measured Hi-C map. The differences in the models we obtain for the wildtype, Δsmc , and rifampicin treatment data sets show that we can use our approach to glean insight into how specific proteins impact chromosome organization.

In principle, it is possible that there is structure that is not fully contained within the Hi-C data of two-point contact frequencies; such structure would only be revealed by three-point or even higher-order contact maps. In this case, the MaxEnt approach presented in our manuscript can be generalized to be constrained simultaneously by two-point and high-order contact maps. However, such experimental data on higher-order contact maps is currently unavailable for *C. crescentus*, and to date there is no evidence that such additional constraints are required to constrain the distribution of single cell conformations.

In this manuscript, we provide an approach to find a distribution of single-cell chromosome conformations with the state-of-the-art two-point Hi-C data at hand. Our approach is based on providing the minimal interpretation of this Hi-C data in terms of a maximum entropy distribution of chromosome conformations. This predicted distribution thus does not contain any structure that is not needed to reproduce the experimental Hi-C map. This has an important implication: if our Hi-C based MaxEnt model makes predictions that fail to describe independent experiments, this would imply that the Hi-C data is not sufficient to fully constrain the distribution of chromosome conformations, which would be an important result in and of itself. Currently, however, we have no strong indications that this is the case.

As explained by the reviewer in their building example, if the measurement does not capture sufficient information to constrain the 3D structures of interest, the resulting MaxEnt model will have limited predictive power. While Hi-C experiments provide strong constraints on the 3D conformations of chromosomes, it is important to test the predictive power of our Hi-C based MaxEnt model. To this end, in our paper, we provide MaxEnt predictions for the long-axis spatial distribution of chromosomal loci, which are in accord with experiments (Fig. 2). In the revised manuscript, we now also provide new predictions of single-cell chromosome density plots, which we compare with new super-resolution data (Fig. 4D,E). We feel that these results (Fig. 2,4) taken together lend credence to our MaxEnt model for the single-cell chromosome conformation in living bacteria. In addition, we provide new predictions of correlation functions (Fig. 3), which can be tested in future experiments.

Regarding the comment on nested loops and condensin. We have started a new project on the role of condensin in *Bacillus subtilis*, to investigate how accurately our MaxEnt approach can capture nested loops and higher-order structures implied by the role of condensin. However, this is still ongoing work on a different bacterium, which is outside the scope of the current manuscript.

Finally, we thank the reviewer for all these constructive comments and suggestions, which helped us improve our manuscript.

Reviewer #1 (Remarks to the Author):

The authors have addressed all my concerns and I find the paper ready for publication.

Two minor (optional) remarks below:

1) Line 211: "the modelled and experimental contact map agree within 6.0%" I am not sure what exactly "agree within 6%" mean, can you please reword to make this more precise?

2) Citation [50] in line 431 should perhaps stand at end of line 422 (maybe parenthetically mention that the concept has been introduced in the context of developmental patterning)?

Reviewer #2 (Remarks to the Author):

The revised manuscript "Learning the distribution of single-cell chromosome conformations in bacteria reveals emergent order across genomic scales" by Messelink et al is significantly improved. Overall, the authors have done a great job and it has turned into a very nice paper! Most of my comments (and those of other reviewers) have largely been addressed. I strongly recommend it for publication at Nature Communications provided that an outstanding issue is resolved.

Major point

- There is a discrepancy in the modified Fig. 2D. The authors claim to have revised the random polymer model to orient the origin position to be in only one cell half (i.e. a distance below 0.5). However, the shown distribution of origin localizations in Fig. 2D has non-zero probability up to 0.6 of the cell length, exceeding the stated imposed maximum of 0.5. Thus, it appears that there is an error in their method of orienting the chromosomes. Accordingly, the "oriented random polymer" average distances in Fig. 1C also exceeds expectation.

Other minor points/suggestions/typos

- In Fig. 1C, the circular chromosome schematic runs contrary to convention; typically, the genome runs clockwise whereas it is shown running counterclockwise.

- Line 307-310: It seems important to elaborate a bit more (perhaps in the supplement) on why SuDs are expected to generate negative intra-arm correlations. It was not initially clear to me why this is expected behavior. Regarding syntax: "negative-anticorrelations" is redundant - I would suggest either "anticorrelations" or "negative correlations".

- Line 344: "along the long axis similar that in wild-type cells" change to "...similar TO THOSE in wild-type..."

- Lines 504-508: it would be appropriate to mention here a paper that directly explores the interplay between SMC complexes and transcription. Brandao et al, PNAS, 2019 show that transcription can

position SMCs and modify SMC-mediated arm juxtaposition in *B. subtilis*. Tran et al 2017 also shows a similar SMC/transcription interplay.

Reviewer #3 (Remarks to the Author):

The authors have addressed most of my comments with great detail. I find that the manuscript has improved significantly, and is now of interest to people without physics background. Given the overall quality of the manuscript and the enthusiasm of other reviewers, I think the manuscript can be published in Nature Communication if the authors address my only remaining comment below.

My only remaining problem with the manuscript is that it does not explain the link between the Hi-C data and SUDs. I was very skeptical about SUDs and remain so. It is very hard for the reader to grasp what "encodes" the information about SUDs in a Hi-C map, and the manuscript forces the readers to blindly trust the result of the MaxEnt model. I find that when such a gap in understanding arises, it is very useful to perform positive and negative controls. The concept of control experiments is very familiar to people with experimental biology background, and will help them understand what stands behind the MaxEnt model, rather than having to trust it as a "black box".

I thought that rifampicin treated cells would be a natural negative control as they would lack SUDs. Indeed, they lack almost any structure in the Hi-C map. Yet they have almost exactly the same number of SUDs per arm. Since there is (virtually) no structure other than P(s) in the rif treated cells, it forces me to believe that MaxEnt approach infers SUDs from the P(s) alone (and maybe the inter-arm diagonal), and not from the features of the Hi-C map such as CIDs.

This realization is not encouraging because P(s) is very experiment-dependent and depends on the protocol differences, such as inSitu vs Dilution (Rao, 2014) or Hi-C vs microC (Krietenstein, 2020). Alarmingly, P(s) tends to be much steeper in new protocols such as inSitu Hi-C or microC, compared to the 6-base-cutter dilution Hi-C used for the *Caulobacter* data. This difference is very often overlooked, and many researchers are not aware of it. For an example of differences between Hi-C and microC see (Krietenstein 2020 Fig 1C) and for dilution-vs-inSitu see Fig 3a in (Comparison of Hi-C results using in-solution versus in-nucleus ligation Nagano et al, Genome Biology 2015).

I encourage the authors to perform a negative control: design synthetic Hi-C data such that a maxEnt reconstruction would have no SUDs or much less SUDs given the same geometry and the same overall structure of the Hi-C map (some P(s); secondary diagonal). Similarly, I would like to see a positive control: what is the minimal set of features in a synthetic Hi-C map required to acquire SUDs? The readers may think that CIDs are, but they are not, because rif treated cells have exactly the same SUDs, but no CIDs. Beyond CIDs there are secondary diagonal and P(s). Is it some particular P(s) that makes SUDs? Or thickness of the secondary diagonal? Or the scaling coefficient between Hi-C and probabilities? It is likely one of these, as there isn't any other information in the rif Hi-C map.

Minor comments:

Reference to S9 on line 337 should be S21? (SUD analysis of rif treated cells)

I agree with the authors about removal of the ori-ter interactions in the reply to the point 2.3: those

are indeed strange, and could be truly considered an artifact. They behave very erratically in Caulobacter Hi-C maps, and sometimes may be different even between biological replicates. The authors' solution to move to unreplicated cells is an even better solution.

REVIEWER COMMENTS

Reviewer #1 (Remarks to the Author):

The authors have addressed all my concerns and I find the paper ready for publication.

Thanks for taking another look, and for these last helpful comments and suggestion.

Two minor (optional) remarks below:

1) Line 211: "the modelled and experimental contact map agree within 6.0%" I am not sure what exactly "agree within 6%" mean, can you please reword to make this more precise?

Our formulation was indeed imprecise, and we now changed this to (page 3, line 209-213):

"Our inverse algorithm robustly converges to an accurate description of the Hi-C map: the modelled and experimental contact maps have an average pair-wise deviation of 6.0% of the total average Hi-C score"

2) Citation [50] in line 431 should perhaps stand at end of line 422 (maybe parenthetically mention that the concept has been introduced in the context of developmental patterning)?

Thanks for this suggestion, which we have now implemented.

Reviewer #2 (Remarks to the Author):

The revised manuscript "Learning the distribution of single-cell chromosome conformations in bacteria reveals emergent order across genomic scales" by Messelink et al is significantly improved. Overall, the authors have done a great job and it has turned into a very nice paper! Most of my comments (and those of other reviewers) have largely been addressed. I strongly recommend it for publication at Nature Communications provided that an outstanding issue is resolved.

Thanks for this careful assessment and for these supportive comments.

Major point

- There is a discrepancy in the modified Fig. 2D. The authors claim to have revised the random polymer model to orient the origin position to be in only one cell half (i.e. a distance below 0.5). However, the shown distribution of origin localizations in Fig. 2D has non-zero probability up to 0.6 of the cell length, exceeding the stated imposed maximum of 0.5. Thus, it appears that there is an error in their method of orienting the chromosomes. Accordingly, the "oriented random polymer" average distances in Fig. 1C also exceeds expectation.

Indeed, the distribution of model positions of the *ori* appears to be non-zero above 0.5. This is a result of the binning procedure we use to plot these results to make a fair comparison between the model and the experimental data. The previously published experimental data shown in Fig. 2B divides the cell along its long axis into 20 bins, where within each bin the corresponding bars for each of the four measured loci are placed next to each other. This causes a locus-dependent shift of the corresponding x-axis values. To make a comparison between model and experiment, we originally plotted the model predictions with the same division into 20 bins, and with the same shift of x-axis values. As the MaxEnt model contains an uneven number of bins along the long axis, the 11th bin of the set of 20 will have a nonzero value.

We thank the reviewer for bringing this to our attention and we see their point: this representation can cause the model predictions to be misread. Therefore, we changed the figure to show all model predictions at the midpoint of their corresponding bin. The experimental results are now displayed to correctly position the average x-coordinate of the bars associated to a specific bin. Additionally, we added the following text to the caption of Figure 2:

"To enable a direct comparison between model and experiment, the model values are distributed over the same number of bins as the experiment."

Regarding the other comment, we believe that the reviewer may be referring to the results in Figure 2A. In this figure, the polymer is oriented such that the origin (at OMB) is always in the lower cell half, and thus has a maximal scaled long-axis position of 0.5. The inferred average long-axis position of ~ 0.38 is consistent with this. We are therefore not sure what the reviewer means with the average distances exceeding expectation. Possibly the reviewer is referring to the average long-axis position of the terminus region, which appears to be slightly above 0.5. This is a deviation due to a finite simulation run time. We ran the corresponding forward simulation for longer, and included the results in the new figure 2A.

Other minor points/suggestions/typos

- In Fig. 1C, the circular chromosome schematic runs contrary to convention; typically, the genome runs clockwise whereas it is shown running counterclockwise.

Thanks for pointing this out. This has been changed now.

- Line 307-310: It seems important to elaborate a bit more (perhaps in the supplement) on why SuDs are expected to generate negative intra-arm correlations. It was not initially clear to me why this is expected behavior. Regarding syntax: "negative-anticorrelations" is redundant - I would suggest either "anticorrelations" or "negative correlations".

We agree that this point could be clarified more. We now included an additional SI figure S22 to illustrate the link between SuDs and the observed anticorrelations.

We modified this sentence to (page 4, line 308-313):

"As a result of this tendency to spatially exclude, chromosomal regions belonging to SuDs on opposing sections of the two arms, are expected to fluctuate in an anti-correlated fashion. (SI Section S9). Thus, this exclusion behavior of opposing SuDs is expected to generate negative intra-arm correlations for pairs of genomic regions with similar average axial positions (SI Section S9)."

Finally, the redundant 'negative anticorrelations' issue has now been resolved.

- Line 344: "along the long axis similar that in wild-type cells" change to "...similar TO THOSE in wild-type..."

Thanks for catching this - this is fixed now.

- Lines 504-508: it would be appropriate to mention here a paper that directly explores the interplay between SMC complexes and transcription. Brandao et al, PNAS, 2019 show that transcription can position SMCs and modify SMC-mediated arm juxtaposition in *B. subtilis*. Tran et al 2017 also shows a similar SMC/transcription interplay.

This is indeed relevant to our observation. We followed the suggestion of the reviewer and included these references towards the end of this paragraph.

Reviewer #3 (Remarks to the Author):

The authors have addressed most of my comments with great detail. I find that the manuscript has improved significantly, and is now of interest to people without physics background. Given the overall quality of the manuscript and the enthusiasm of other reviewers, I think the manuscript can be published in Nature Communication if the authors address my only remaining comment below.

We appreciate the reviewer's supportive comments and for their remaining thoughtful comments/suggestions.

My only remaining problem with the manuscript is that it does not explain the link between the Hi-C data and SUDs. I was very skeptical about SUDs and remain so. It is very hard for the reader to grasp what "encodes" the information about SUDs in a Hi-C map, and the manuscript forces the readers to blindly trust the result of the MaxEnt model. I find that when such a gap in understanding arises, it is very useful to perform positive and negative controls. The concept of control experiments is very familiar to people with experimental biology background, and will help them understand what stands behind the MaxEnt model, rather than having to trust it as a "black box".

I thought that rifampicin treated cells would be a natural negative control as they would lack SUDs. Indeed, they lack almost any structure in the Hi-C map. Yet they have almost exactly the same number of SUDs per arm. Since there is (virtually) no structure other than P(s) in the rif treated cells, it forces me to believe that MaxEnt approach infers SUDs from the P(s) alone (and maybe the inter-arm diagonal), and not from the features of the Hi-C map such as CIDs.

This realization is not encouraging because P(s) is very experiment-dependent and depends on the protocol differences, such as inSitu vs Dilution (Rao, 2014) or Hi-C vs microC (Krietenstein, 2020). Alarmingly, P(s) tends to be much steeper in new protocols such as inSitu Hi-C or microC, compared to the 6-base-cutter dilution Hi-C used for the *Caulobacter* data. This difference is very often overlooked, and many researchers are not aware of it. For an example of differences between Hi-C and microC see (Krietenstein 2020 Fig 1C) and for dilution-vs-inSitu see Fig 3a in (Comparison of Hi-C results using in-solution versus in-nucleus ligation Nagano et al, Genome Biology 2015).

I encourage the authors to perform a negative control: design synthetic Hi-C data such that a maxEnt reconstruction would have no SUDs or much less SUDs given the same geometry and the same overall structure of the Hi-C map

(some P(s); secondary diagonal). Similarly, I would like to see a positive control: what is the minimal set of features in a synthetic Hi-C map required to acquire SUDs? The readers may think that CIDs are, but they are not, because rif treated cells have exactly the same SUDs, but no CIDs. Beyond CIDs there are secondary diagonal and P(s). Is it some particular P(s) that makes SUDs? Or thickness of the secondary diagonal? Or the scaling coefficient between Hi-C and probabilities? It is likely one of these, as there isn't any other information in the rif Hi-C map.

We thank the reviewer for the suggestion of analyzing the connection between Hi-C map features and the presence of SuDs. This is a very interesting question, which is challenging to answer rigorously and generally. A complete answer to this question would require a systematic sweep over a large number of possible Hi-C maps, which is unfortunately computationally unfeasible (For each specific map, we need to perform an inverse algorithm to infer a MaxEnt model, followed by a forward simulation of the converged model – all of which takes about half a week with our current computational recourses). We did consider the reviewers' suggestion to consider a few artificial Hi-C maps. However, artificially generated Hi-C maps might yield unphysical scenarios, and we therefore find it difficult to interpret the results of analyzing MaxEnt models on artificial Hi-C maps.

Inspired by the suggestion of the reviewer, we realized that we can gain insight into the link between features of Hi-C maps and SuDs by considering which features of existing (not artificial) Hi-C maps are necessary for the presence of SuDs. The key idea is that we can also learn a MaxEnt model on a subset of the Hi-C map, leaving the remaining Hi-C scores as unknown. Thus, we ran versions of the MaxEnt model trained on a subset of Hi-C scores for rifampicin-treated cells (for which the CIDs are indeed already absent). We analyzed the following subsets of Hi-C data:

- 1) Only short-distance Hi-C scores. This encodes P(s) for short distances, and no further information. We included Hi-C scores for regions with a genomic distance of up to 0.4 Mb (Figure R1 A).
- 2) Only long-distance Hi-C scores. This is the inverse of case 1). We included Hi-C scores for regions with a genomic distance of more than 0.4 Mb (Figure R1 B).
- 3) Only the second diagonal. We included Hi-C scores for regions on opposite chromosomal arms with a genomic distance of more than 0.4 Mb, and a distance to the second diagonal of less than 0.6 Mb (Figure R1 C).
- 4) For comparison, we also included results for the oriented random polymer, which corresponds to not imposing any Hi-C constraints (Figure R1 D).

We find that of these four scenarios, only the long-distance constraints (case 2)) yield SuDs (Figure R1 F). This implies that information on P(s) for short distances is not necessary for the presence of SuDs. Moreover, only having information on the second diagonal is not sufficient to yield SuDs (Figure R1 G). While SuDs still appear for the long-distance constraint case, their properties are different compared to the results for the full data set. The average SuD size is reduced by 4% compared to the full data set, and the exclusion between SuDs is much weaker (1% less overlap between SuD centers compared to randomly paired arms, compare with the 24% for the full data set). This implies that although the information on P(s) for short distances is not needed for SuDs to be present, this information can affect SuD properties.

Analyzing localization patterns for these four cases, we find that only the long-distance Hi-C constraints (case 2)) yield a linear organization similar to results for the full Hi-C data set (Figure R1 J). This suggests a link between SuD presence and a linear chromosome organization. The results of these analyses yield useful insights into SuD origins and properties. However, we don't think these results are necessary to support the conclusions we make in this manuscript, which is already rather long and dense. Instead, we feel these results would be done more justice in our planned follow-up work diving deeper into SuD properties, and we would therefore prefer to include them therein.

With regards to the concern raised by the reviewer on the role of P(s), these results demonstrate that knowledge of P(s) is not necessary to produce SuDs. Additionally, we see that P(s) varies between different data sets yielding SuDs that we consider (Figure R2). This rules out that the SuDs could purely be the result of one particular form of P(s).

The reviewer further raises the point that certain features of the Hi-C data sets could be artefacts due to the experimental procedure used. The data sets analyzed here represent the state-of-the-art on bacterial chromosomes, making it the most suitable data set to apply our method to. If new data sets become available, it would be very interesting to apply the MaxEnt procedure to these and compare results. We will make our code publicly available to facilitate this.

Thus, we find that the emergence of SuDs does not appear to be caused by one very specific feature of the Hi-C map, which may be sensitive to the experimental protocol such as the P(s) scaling, but rather by the overall structure of the long-ranged contacts. In addition, we emphasize that SuDs appear robustly for a range of conditions (Δsmc , rifampicin, with and without data processing, replication inhibited cells). Taken together, we hope that these arguments address the remaining concerns of the reviewer.

Figure R1 Analysis of SuD presence for MaxEnt models trained on subsets of the full Hi-C data set for rifampicin-treated cells. A) Input contact frequencies (upper left) and resulting model contact frequencies (lower right) if only constraints are incorporated for genomic regions less than 0.4 Mb apart. **B)** Similar to A), now with Hi-C constraints incorporated for genomic regions more than 0.4 Mb apart. **C)** with Hi-C constraints incorporated for genomic regions more than 0.4 Mb apart, and a distance to the second diagonal axis of less than 0.6 Mb. **D)** With no Hi-C constraints incorporated (oriented random polymer). **E-H)** Cluster scaling analysis for each of the input data sets. Only case 2) shown in panel B) shows a typical length scale at which clusters are formed, indicating the presence of SuDs. **I-L)** Average long-axis positions for each of the input data sets. Only condition B) displays a linear organization along the long axis similar to a model trained on the full Hi-C data set.

Figure R2 Scaling of average contact frequencies with inter-arm genomic distance for four data sets that yield SuDs. The scaling behavior for genomic distances up to 0.37Mb are shown. The long-distance constraint for rifampicin treated cells is derived from the contact frequencies shown in Figure R1B (lower right).

Minor comments:

Reference to S9 on line 337 should be S21? (SUD analysis of rif treated cells)

The intended reference was to SI Section S9. To avoid ambiguity, all references to SI sections are now denoted by 'SI Section SX'.

I agree with the authors about removal of the ori-ter interactions in the reply to the point 2.3: those are indeed strange, and could be truly considered an artifact. They behave very erratically in Caulobacter Hi-C maps, and sometimes may be different even between biological replicates. The authors' solution to move to unreplicated cells is an even better solution.

Thanks for this comment.

Reviewer #2 (Remarks to the Author):

The authors have done a great job addressing all of my outstanding concerns. I strongly recommend for publication at Nature Communications.

Reviewer #3 (Remarks to the Author):

The authors have addressed all of my comments.

I'm somewhat surprised by such a long runtime of the algorithm. If it's so slow, then the performance evaluation and the maximum size of the system it can be realistically applied to, should be surely noted in the text and/or on github.